# RULEREASONER: REINFORCED RULE-BASED REASONING VIA DOMAIN-AWARE DYNAMIC SAMPLING

**Yang Liu**[⋆]    **Jiaqi Li**[⋆]    **Zilong Zheng**[†]
State Key Laboratory of General Artificial Intelligence, BIGAI
{liuyang,lijiaqi,zlzheng}@bigai.ai

## ABSTRACT

Rule-based reasoning is acknowledged as one of the fundamental problems of reasoning. While recent studies show that large reasoning models (LRMs) have remarkable reasoning capabilities enhanced by reinforcement learning (RL), real applications still face severe challenges due to variations in rule formats, types, and complexity. To mitigate this issue, we introduce **RULEREASONER**, an effective method for rule-based reasoning via a wide collection of curated tasks and a novel domain-aware dynamic sampling approach in RL. Specifically, RULEREASONER resamples each training batch by updating the domain weights based on historical rewards. This facilitates domain balance and active learning schedules for RL, obviating static mix-training engineered by humans. Evaluations of in-distribution (ID) and out-of-distribution (OOD) benchmarks reveal that RULEREASONER outperforms frontier LRMs by a significant margin ($\Delta$**4.1%** on eight ID tasks and $\Delta$**10.4%** on three OOD benchmarks over OpenAI-o1). Notably, our approach also exhibits higher computational efficiency compared to prior methods.

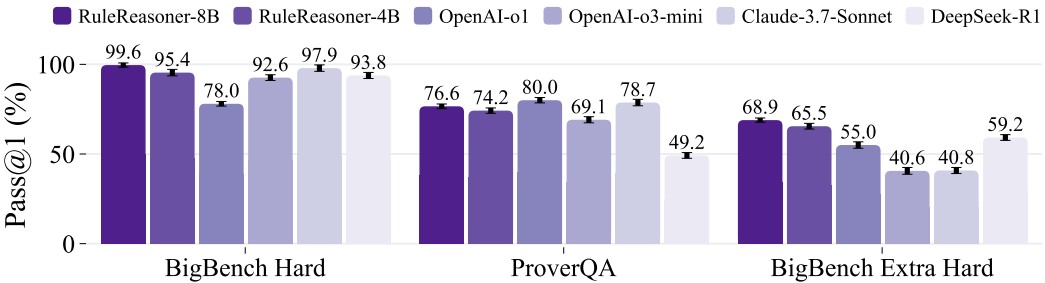

Figure 1: Out-of-distribution performance comparison between RULEREASONER (8B and 4B) and other frontier reasoning models on challenging rule-based reasoning benchmarks.

## 1 INTRODUCTION

Rule-based reasoning (Xu et al., 2024b; Wang et al., 2024c; Servantez et al., 2024b; Morishita et al., 2024; Wang et al., 2024d) is an ability to draw new conclusions or make decisions based on provided facts and predefined logical rules, which requires a strong ability of reasoning. It emulates human reasoning and mirrors the structured deductive processes that humans employ in domains such as law, finance, and medical diagnostics (Liu et al., 2023b; Xiong et al., 2024; He et al., 2025b). The need for rule-based reasoning increasingly grows in scenarios that require transparency, explainability, and adherence to domain constraints. Moreover, deviations from rules for different scenarios lead to significant changes in reasoning process, which requires more controllable and adaptable reasoning capabilities under ever-changing circumstances (Saparov et al., 2023; Tang et al., 2023a; 2024; Li et al., 2025b).

Recent work has demonstrated the remarkable reasoning capabilities of large reasoning models (LRMs) with an intermediate thinking process, chain-of-thought (CoT) (Wei et al., 2022b), notably

---

[⋆]Equal contribution. [†]Corresponding author.

the long thinking ability stimulated by reinforcement learning (RL) (Xie et al., 2025; Meng et al., 2025; Guo et al., 2025; Hu et al., 2025). However, conventional approaches rely closely on scaling to larger model sizes or supervision distilled from superior models. What's more, as the contextual window expands, language models encounter difficulties like *"lost in the middle"* (Liu et al., 2024; An et al., 2024) that leads to performance collapse in long-dependency reasoning tasks (Li et al., 2024; Mao et al., 2025). The limit exhibits relatively weak instruction following abilities to understand and associate rules and facts provided in the context (Srivastava et al., 2025; Li et al., 2025d; Guan et al., 2025) required for task completion.

With this in mind, we aim at investigating **whether it is feasible and effective to enhance the rule-based reasoning ability of language models**. We also want to validate that this improved capability can generalize to various and unseen reasoning tasks, forms, and difficulties. Therefore, we propose **RULEREASONER**, which trains language models to be rule-based reasoners. It performs on-policy RL and mitigates its limitations of sample inefficiency and the application of rigid rules in dynamic contexts. Specifically, RULEREASONER leverages task reward to estimate domain sampling weights without requiring human prior knowledge or high computational costs from repeated rollouts. First, RULEREASONER initializes to train a model in a standard Reinforcement Learning with Verifiable Rewards (RLVR) way. Second, within a training iteration, RULEREASONER updates the domain weights based on historical rewards of previous training steps. Finally, RULEREASONER samples a training batch based on the domain weights to perform policy optimization.

**Our innovations** are summarized as three folds:

- **Novel Rule-centric Dataset:** We release a large and diverse dataset, RULECOLLECTION-32K, which spans eight rule-based reasoning tasks with explicit or implicit contextual rules tied to each question. These rules vary in format, reasoning forms, and complexity, allowing training and ID/OOD evaluation in generalizable rule application rather than memorization.

- **RLVR for Rule-based Reasoning:** We design a RLVR framework that introduces useful training regularization to achieve stable RL training dynamics on complex rules, even with model size under 8B. It encourages exploring and exploiting valid reasoning steps instead of imitation, improving generalization to unseen rules in the RULECOLLECTION-32K.

- **Domain-aware Dynamic Sampling:** To harmonize language models' proficiencies across imbalanced domains, we present an adaptive sampling algorithm that dynamically reweights training domains based on their degree of under-optimization. This ensures balanced learning dynamics across tasks, enhancing both in-distribution (ID) and out-of-distribution (OOD) performance.

We perform extensive evaluations on RULEREASONER, introducing two best rule-based reasoning models: RULEREASONER-8B and RULEREASONER-4B. Empirical results show that: (1) As depicted in Figure 1, RULEREASONER-8B outperforms OpenAI-o1, Claude-3.7-Sonnet, and DeekSeek-R1, achieving higher performance than trivial RLVR methods. Specifically, RULEREASONER-8B achieves $\Delta 14\%$ and $\Delta 49\%$ OOD pass@1 respectively over o1 and the base model. (2) RULEREASONER-4B further demonstrates that language models can effectively learn rules even with a smaller model size, achieving an average pass@1 of 78.3% on three OOD benchmarks ($\Delta 7.3\%$ over o1). (3) Surprisingly, RULEREASONER achieves comparable task performance with notably fewer training steps than existing RLVR methods. This suggests that RULEREASONER not only enhances ID and OOD performance but also improves training efficiency.

## 2 RELATED WORK AND PRELIMINARIES

RL plays a critical role in improving the reasoning capabilities of large language models (LLMs) (Silver & Sutton, 2025), particularly through approaches such as RLVR (Guo et al., 2025; Li et al., 2025c; Zuo et al., 2025). In this section, we introduce the task definition of rule-based reasoning (§2.2) and discuss key components of prior RLVR methods (§2.3) and their limitations.

### 2.1 RELATED WORK

It has been demonstrated that LLMs can learn and apply rules, with promising potential (Zhu et al., 2023; Servantez et al., 2024a). Early works start from predefined rules in symbolic forms with an emphasis on scalability and compositionality in specific tasks (Tang et al., 2023b; Luo et al., 2024;

Jia et al., 2024; Gui et al., 2024; He et al., 2025a) while recent works are dedicated to perform rule-based reasoning in natural language that are more applicable for real scenarios (Zhou et al., 2024; He et al., 2024; Tang et al., 2024). It is also worth noting that recent advances in logical reasoning, such as Logic-RL (Xie et al., 2025), are generally considered rule-free for reasoning, which differs from our task definition. These methods explore the potential of rule learning through diverse prompting methods (Diallo et al., 2025; Peng et al., 2024), supervised distillation (Wang et al., 2024a), and external memory augmentation (Wang et al., 2024b;d). However, they spend less effort adapting the reasoning capability of LLMs to unseen tasks with limited task types and formats. Inspired by recent advancements in RLVR methods focused on mathematical reasoning and code generation (Zhang et al., 2025; Chen et al., 2025; Wei et al., 2025; Li et al., 2025b; Zhao et al., 2025; Li et al., 2025a), we further optimize their limitation on data efficiency with dynamic data sampling along with a curated collection of diverse rule-centric training data. Our method improves model performance across both ID and OOD reasoning tasks with higher generalization and computational efficiency. Conventional reweighting and loss-based sampling methods (Luo et al., 2025; Shi et al., 2025b) are unsuitable for RLVR because the direction of optimization for surrogate loss of RL does not correlate with policy model performance. Moreover, compared with prior methods, we unifies domain-level dynamic sampling strategy into RL for reasoning without requiring proxy models to obtain domain weights (Xie et al., 2023; Liu et al., 2025a), instead leveraging only the verifiable rewards in a self-adaptive paradigm.

## 2.2 PRELIMINARIES I: RULE-BASED REASONING

Given a question, a set of **facts**, and associated **rules** as context, the model is asked to answer the question by applying and reasoning the facts with rules. In this paper, we refer to the rules as *contextual logic rules*, which are expressed in natural language and given specifically for each question. The rules provided can be generated either explicitly or implicitly as principles or premises to solve the question. For a grouped datasets $\mathcal{D} := \{(d_i, q_i, r_i, y_i)\}_{i=1}^n$, where $d_i \in \{d_1, \cdots, d_n\}$ is a specific domain, $q_i$ is a question, $r_i$ is a reasoning trajectory, and $y_i$ is a verifiable answer.

## 2.3 PRELIMINARIES II: ON-POLICY REINFORCEMENT LEARNING

**Reward Shaping.** To teach models to learn reasoning, we design a rule-based exact match (EM) reward function to evaluate the response according to the final answer, ensuring both the correctness of the answers and the adherence to the format. We define $(q, \hat{y}) \sim \mathcal{D}$, $y \sim \pi_\theta(\cdot|q)$, and

$$\mathcal{R}_{\text{EM}}(\hat{y}, y) = \begin{cases} 1 & \text{is\_equivalent}(\hat{y}, y), \\ -1 & \text{otherwise}. \end{cases} \tag{1}$$

**Policy Optimization.** We adopt the basic form of GRPO (Shao et al., 2024) but discard the part of KL loss, encouraging the model to explore various solutions. For each question-answer pair $(q, y)$, policy model $\pi_{\theta_{\text{old}}}$ samples a group of responses $\{y_1, y_2, \ldots, y_G\}$ and calculates the associated rewards $\{r_1, r_2, \ldots, r_G\}$, given the oracle answer $y$, using the aforementioned reward function $\mathcal{R}_{\text{EM}}$.

$$\mathcal{J}(\theta) = \mathbb{E}_{(q,y)\sim\mathcal{D}, \{y_i\}_{i=1}^G \sim \pi_{\theta_{\text{old}}}(\cdot|q)}$$
$$\left[ \frac{1}{G} \sum_{i=1}^G \frac{1}{|y_i|} \sum_{t=1}^{|y_i|} \left( \min\left( r_{i,t}(\theta) A_{i,t}, \text{clip}\left( r_{i,t}(\theta), 1-\varepsilon, 1+\varepsilon \right) A_{i,t} \right) \right) \right], \tag{2}$$

where $r_{i,t}(\theta)$ is the rate of importance sampling for domain $d_i$ at the $t$-th token for $y_i$, and $A_i$ is the advantage as the critic obtained by normalizing the rewards within each group. We strictly follow the on-policy training method, performing only one gradient update after the policy model $\pi_{\theta_{\text{old}}}$ generates a group of $G$ rollouts, to enable stable RL training and prevent entropy collapse.

$$r_{i,t}(\theta) = \frac{\pi_\theta(y_{i,t} \mid q, y_{i,<t})}{\pi_{\theta_{\text{old}}}(y_{i,t} \mid q, y_{i,<t})}, \quad A_i := \widetilde{r_i} = \frac{r_i - \text{mean}\left(\{r_1, r_2, \cdots, r_G\}\right)}{\text{std}\left(\{r_1, r_2, \cdots, r_G\}\right)}. \tag{3}$$

**Limitations of RLVR on Training Efficiency.** Though current RLVR elicits the long chain-of-thought reasoning ability based on the policy gradient RL algorithm like PPO (Schulman et al.,

2017) and GRPO (Shao et al., 2024), the efficiency of training data for RLVR remains relatively unexplored. Existing works like DAPO (Yu et al., 2025) oversamples and filters out prompts with the accuracy equal to 1 and 0 to enhance training efficiency. However, it does not push the limits of training efficiency due the large recompute cost in the rollout stage. SRPO (Zhang et al., 2025b) shows the gains via epoch-level re-sampling wtih RLVR without exploring the agile sampling methods for fine-grain control. Moreover, ADARFT (Shi et al., 2025a) explores an efficient batch-level sampling method using curriculum learning; however, it relies on human priors or an empirical success rate by models on sample difficulty. In the following sections, we expend great effort to further leverage training examples to achieve higher reasoning performance effectively.

# 3 DOMAIN-AWARE POLICY OPTIMIZATION WITH DYNAMIC SAMPLING

In data sampling, increasing the number of samples in an individual domain potentially harms the performance of other domains without timely control (Albalak et al., 2024) or causes obvious tradeoff across domains (Xie et al., 2023).

To address similar issues in RLVR, as shown in Alg. 1 and Figure 2, we propose **D**omain-**a**ware **D**ynamic **S**ampling (DADS), an effective sampling method for RLVR aiming to improve the performance of a policy $\pi_\theta$ for solving multi-domain rule-based reasoning tasks. DADS dynamically adjusts the probability of sampling

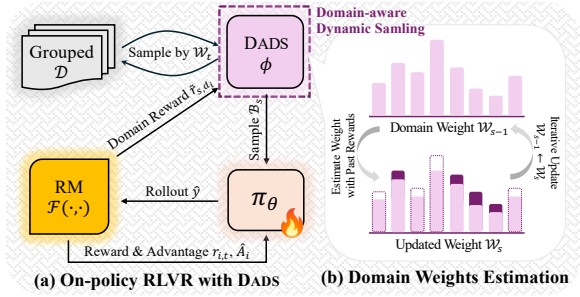

Figure 2: RULEREASONER training recipe.

data from different domains based on their historical rewards. By prioritizing domains that yield lower verifiable rewards or those lagging behind a target reward, DADS enhances sample efficiency in training batch $\mathcal{B}_s$ and mitigates the domain imbalance issue, leading to faster and more stable learning of policies that satisfy reward specifications. We instantiate RULEREASONER with the gradient policy algorithm of GRPO variant in this work to demonstrate its effectiveness and efficiency.

## 3.1 DOMAIN-AWARE DYNAMIC SAMPLING (DADS)

**Domain-aware Rewards.** At each training step $s$, to evaluate the proficiency for a domain $d_i \in \mathcal{D}$, we define $\bar{r}_{d_i,s}$ as the algebraic mean, calculated by domain rewards $\{r_{s,d_i,j}\}_{j=1}^m \sim \mathcal{R}_{\text{EM}}(y, \hat{y})$ of $m$ previous samples in the domain, which correspond to genersations $\mathcal{Y}_{s-1} : \{y_{s,d_i,j}\}_{j=1}^m \sim \pi_\theta(\cdot|q)$ and the set of ground truth $\hat{\mathcal{Y}}_{s-1}$. Note that $m$ may vary across different domains and training steps due to the batch-level domain sampling strategy. Domain-aware rewards calculation over batch (Alg. 1, line 6 and 7) is computed as: $\bar{r}_{s,d_i} = \frac{1}{m}\sum_{j=1}^m r_{s,d_i,j}$. We employ a target reward, $r_{\text{target}} \triangleq 1$, to define the upper bound for the underoptimization estimation, $v_{s,d_i}$, of a domain. Thus, we have the estimation: $1 - \widetilde{r}_{s,d_i}$, where target reward quantifies the extent to which a domain performance lags behind the desired level.

**Decaying Importance Sampling.** Furthermore, given the utilization of past rewards for domain $d_i$, we introduce a decaying importance-sampling strategy, which employs the exponentially weighted moving average (Holt, 2004) that considers both current and the historical estimated rewards. The historical rewards $\{\widetilde{r}_{s-1,d_i}\}_{i=1}^n$ are involved with the smoothing factor $\alpha$ to produce normalized rewards $\{\widetilde{r}_{s,d_i}\}_{i=1}^n$. We have $\widetilde{r}_{s,d_i} = \alpha\widetilde{r}_{s-1,d_i} + (1-\alpha)\bar{r}_{s,d_i}$, where $\alpha \in [0,1]$ serves as a smoothing factor that creates a more stable estimate of the performance for a domain over time, rather than relying solely on the most recent reward $\bar{r}_{s,d_i}$.

**Domain Re-weighting.** Consequently, we establish a domain weight, $w_{s,d_i}$, which is then normalized using a standard softmax function (as detailed in Alg. 1, lines 11 to 13). In this normalization, hyperparameters $\tau$ and $\epsilon$ are used: $\epsilon$ ensures a minimum sampling weight for all domains, even well-learned ones, and $\tau$ adjusts how strongly the sampling prioritizes domains based on their rewards. We discuss the sensitivity of $\alpha$, $\tau$, and $\epsilon$ in §C.5. After obtaining the re-sampling weights

$\mathcal{W}_s := \{w_1, w_2, \ldots, w_n\}$ across domains, we use $\mathcal{W}_s$ to construct a new batch $\mathcal{B}_s$ for the subsequent policy optimization iteration. The process will be implemented iteratively during the training and more details are described in Algorithm 1.

## 3.2 TRAINING REGULARIZATION

We want to ensure that the model follows, matches, and implements the rules and does not just learn to identify specific datasets and perform correctly on them. To do this, we introduce different kinds of noise during RL training. This stops the models from recognizing particular datasets, recalling specific rules, or memorizing only the similar seen facts in the context.

**Disabling Entropy Bonus.** We discard the entropy bonus employed by Schulman et al. (2017) and Shao et al. (2024), to avoid the issue of entropy explosion in RL without the coldstart bootstrap.

**Discarding KL Divergence.** Similary to Liu et al. (2025b), we eliminate the KL term since the concerns on distributional shift of $\pi_\theta$ is eliminated by rule-based reward function defined in Eq. 1. This saves memory and computation in training while encouraging more exploration required by $\pi_\theta$.

**Rules Order Shuffling.** To prevent memorization of ordered rules appeared in RULECOLLECTION-32K, the order of contextual logical rules are randomly shuffled for each training sample.

---

**Algorithm 1** Domain-aware Dynamic Sampling

**Input:** Policy model: $\pi_\theta : \mathcal{X} \to \mathcal{Y}$;
        Reward model: $\mathcal{R}_{\text{EM}}(\cdot, \cdot) : \mathcal{Y}, \hat{\mathcal{Y}} \to \{0, 1\}$;
        Last weight: $\mathcal{W}_{s-1} := \{w_1, w_2, \ldots, w_n\}$;
        Grouped data: $\mathcal{D} := \{(d_i, q_i, r_i, y_i)\}_{i=1}^n$
        where domain: $d_i \in \{d_1, \cdots, d_n\}$;
        Hyperparameters: $\{\alpha, \epsilon, \tau\} \subset \mathbb{R}^+$.
**Output:** Constructed batch of samples: $\mathcal{B}_s$.
1: **procedure** TRAIN STEP $s$ SAMPLING
2:     **Initialize:** $\mathcal{B}_{s-1} \leftarrow \mathcal{W}_{s-1} \times \mathcal{D}$; $\tilde{r}_{0,d_i} \leftarrow 0$.
3:     $\mathcal{Y}_{s-1} \leftarrow \pi_\theta(\mathcal{B}_{s-1})$     ▷ **ROLLOUT**
4:     $\{\{r_{s,d_i,j}\}_{j=1}^{m_i}\}_{i=1}^n \leftarrow \mathcal{R}_{\text{EM}}(\mathcal{Y}_{s-1}, \hat{\mathcal{Y}}_{s-1})$
5:     /* Update estimated rewards */
6:     $\{\bar{r}_{s,d_i}\}_{i=1}^n \leftarrow \{\frac{1}{m_i} \sum_{j=1}^{m_i} r_{s,d_i,j}\}_{i=1}^n$
7:     $\{\tilde{r}_{s,d_i}\}_{i=1}^n \leftarrow \{\alpha \tilde{r}_{s-1,d_i} + (1-\alpha)\bar{r}_{s,d_i}\}_{i=1}^n$
8:     /* Calculate weights by rewards */
9:     **for** $i = 1, 2, \ldots, n$ **do**
10:         $v_{s,d_i} \leftarrow 1 - \tilde{r}_{s,d_i}$
11:         $w_{s,d_i} \leftarrow \exp(v_{s,d_i}/\tau) + \epsilon$
12:     $\mathcal{W}_s := \{w_{s,d_i}^{\text{norm}}\}_{i=1}^n$   ▷ **NORMALIZING**
13:         $= \{w_{s,d_i}/\sum_{j=1}^n w_{s,d_j}\}_{i=1}^n$
14:     /* Re-sample w.r.t. optimized weights */
15:     $\mathcal{B}_s \leftarrow \mathcal{W}_s \times \mathcal{D}$   ▷ **SAMPLING BY $\mathcal{W}_s$**
16:     **return** $\mathcal{B}_s$

---

## 3.3 RULECOLLECTION-32K: LOGICAL RULES DATA CURATION

We follow the following **principles** to collect our training data, namely RULECOLLECTION-32K.

- **Varying Depths.** We collect 0-7 hop reasoning data for curriculum learning (Bengio et al., 2009) across complexity levels and forms (deductive, inductive, analytical).
- **Different Formats.** Collected data includes explicit or implicit rules as premises or constraints, enhancing the model's flexibility in recognizing, parsing, and applying rules in diverse contexts.
- **Multiple Inference Rules.** We focus the diversity of rules of inference, which implies in the dataset facilitates learning dynamic rule employment and length generalization further.
- **Context Dependency.** We focus on applying contextual rules adaptively for different questions. Correctly answering requires more than memorizing rules, reasoning, or using common sense.
- **Robust Evaluation.** We prioritize boolean and multiple-choice questions over free text to make it more conducive to obtaining rule-based outcome rewards and precise evaluation results.

These principles are critical to ensure training data captures the complexity and diversity inherent in rule-based reasoning tasks. The statistics of training data are presented in Table 5. **We highlight the importance and necessity of RULECOLLECTION-32K with further analyses in §C.9.**

## 4 EXPERIMENTS

### 4.1 EXPERIMENTAL SETUP

| Task | Context (Explicit or Implicit Rules) | Question | Answer |
|---|---|---|---|
| **ProofWriter** | **RULES:** *If the bear needs the dog and the dog visits the bear then the bear likes the cat. If something is rough then it likes the dog* **FACTS:** *The bear is round. The bear visits the catc* | The bear needs the cat? | True |
| **ProntoQA** | **RULES:** *Everything that is earthy and a wumpus is an impus. Everything that is dull and a brimpus is a numpus* **FACTS:** *Sally is dull. Sally is a brimpus.* | Sally is dull and a brimpus? | True |
| **Clutrr** | **RULES:** *If B is the son of A, and C is the grandmother of B, then C is the mother of A.* **FACTS:** *Pedro is taking his wife Dorothy out to dinner for their date tonight. Tracy loves cooking for her son. Tracy went to the store with her sister Dorothy.* | How is Shantel related to Pedro? | Shantel is the mother -in-law of Pedro. |
| **LogicNLI** | **RULES:** *All not fierce people are not brainy. If there is at least one people who is not intelligent, then Keaton is fragile and Jaime is fierce.* **FACTS:** *Jaime is fragile. Philip is not sociable. Jaime is brainy.* | Landon is not intelligent. | Entailment |
| **FOLIO** | *Rafa Nadal was born in Mallorca. Rafa Nadal is a professional tennis player. Nadal's win ratio is higher than 80%. All players in the Big 3 are professionals who have a high win ratio.* | Nadal was not born in Mallorca. | False |
| **Logical Deduction** | *On a shelf, there are five books: a blue book, a red book, a purple book, a gray book, and a white book. The white book is to the right of the gray book. The blue book is the leftmost. The red book is to the left of the gray book. The red book is the third from the left.* | Which of the following is true? A) The blue book is the second from the right. B) ... C) ... | D |
| **AR-LSAT** | *Eight new students—R, S, T, V, W, X, Y, Z—are being divided among exactly three classes—class 1, class 2, and class 3. Classes 1 and 2 will gain three new students each; class 3 will gain two new students.* | If T is added to class 3, which one of the following is a student who must be added to class 2? | C |
| **LogiQA** | *Xiao Ming forgot what day it was today, so he asked O, P, and Q. O replied I also forgot what day it is today, but you can ask P and Q both. P replied Yesterday It's the day when I lied. Q's answer is the same as P. It is known that 1.O never lied;* | What day is today? A) Monday B) Tuseday C) Thursday D) Sunday | C |

Figure 3: Demonstration overview of RULECOLLECTION-32K.

**Datasets and Benchmarks.** We assess the generalization of models on unseen tasks using subsets from BigBench Hard (Suzgun et al., 2023), BigBench Extra Hard (Kazemi et al., 2025), and ProverQA (Qi et al., 2025), as detailed in Table 1. We also employ AIME 2025 (AIME, 2025), GPQA (Diamond) (Rein et al., 2024), and Coin Flip (Wei et al., 2022b) as additional evaluation to assess

Table 1: OOD benchmarks statistics.

| OOD Test | Examples | Levels |
|---|---|---|
| BBH | 750 | ✓ |
| BBEH | 400 | ✓ |
| ProverQA | 1,500 | ✓ |

whether RULEREASONER can extrapolate to more general reasoning tasks. More details on RULECOLLECTION-32K (Figure 3) are presented in §B.1.

**Compared Baselines.** We include five types of baselines: **(1) Prior rule-based reasoners (RBRs):** Hypotheses-to-Theories (Zhu et al., 2023), Logic-of-Rule (Servantez et al., 2024a), and Rule-Guided Feedback (Diallo et al., 2025); **(2) Frontier reasoners:** OpenAI-o1 (*o1-2024-12-17*) (Jaech et al., 2024), o3-mini (*o3-mini-2025-01-31*) (Zhang et al., 2025a), DeepSeek-R1 (Guo et al., 2025), and Claude-3.7-Sonnet (*claude-3-7-sonnet-20250219* with thinking mode) (Anthropic, 2025) with standard zero-shot CoT prompting (Wei et al., 2022b); **(3) Behavioral cloning** (Pomerleau, 1988)[1]: SFT without CoT (Wei et al., 2022a), SFT with short CoT (Yeo et al., 2025), and SFT with distilled long CoT (Yeo et al., 2025) from o3-mini; **(4) Advanced RLVRs:** we compare RLVR approaches including GRPO (Shao et al., 2024), Dr. GRPO (Liu et al., 2025b), and DAPO (Yu et al., 2025) ; **(5) Curriculum Learning:** we also introduce recent curriculum learning approaches for RL: ADARFT (Shi et al., 2025a), and data-balance and easy-to-hard strategies from Parashar et al. (2025).

**Evaluation Metrics.** All tasks in the work are evaluated using the algebraic mean of hard exact match, which is also equivalent to pass@1 accuracy under strict extraction and comparison.

**Implementation Details. Training Setup:** We use Qwen3 (4B and 8B base) (Yang et al., 2025b) as the base models and employ veRL (Sheng et al., 2024) for RL post-training. We set train and mini batch sizes to 64 for strict on-policy updates, and a rollout size of 64 per question. For hyperparameters in DADS, we use a $\tau$ of 0.5 for moderately frequent domain weight updates and an $\epsilon$ of 0.1 for minimum sampling probability per domain, with a smoothing factor $\alpha$ of 0.5. **Inference Setup:** We employ random sampling (temperature $\tau = 0.6$ and *top-p* $= 0.95$ with a maximum output length of 2,048. For the rest of baselines, we use Qwen3-8B-Base for full-parameter SFT or RL. We perform five runs per test set and report the mean and standard deviation of the performance.

## 4.2 RULEREASONER IMPROVES RLVR PERFORMANCE AND EFFICIENCY

**In-Distribution Performance.** As shown in Table 2, we report the ID tasks performance to depict the effectiveness of RULEREASONER. Compared with the frontier LRMs, RULEREASONER-

---

[1]Following RL literature nomenclature, we refer to models trained with the negative log-likelihood loss as behavioral cloning and perform task-focused supervised training to maximize baseline performance.

Table 2: Comparison with all baselines on eight ID benchmarks. RULEREASONER significantly outperforms most of other methods. Average is the macro mean across all samples of domains.

| | Induction | Deduction | | FOL | | Others | | | Avg. Results |
|---|---|---|---|---|---|---|---|---|---|
| | Clutrr | ProntoQA | ProofWriter | FOLIO | LogicNLI | AR-LSAT | Logic. Dedu. | LogiQA | |
| **PRIOR RBRs** | | | | | | | | | |
| HtT (Zhu et al., 2023) | 40.3 | 92.0 | 88.0 | 71.0 | 54.0 | 97.0 | 100.0 | 79.1 | 77.7 |
| RGFB (Diallo et al., 2025) | 31.3 | 94.0 | 88.0 | 74.0 | 55.0 | 95.0 | 100.0 | 79.1 | 77.1 |
| Chain-of-Logic (Servantez et al., 2024a) | 44.8 | 91.0 | 92.0 | 80.0 | 54.0 | 97.0 | 100.0 | 80.6 | 80.0 |
| **FRONTIER REASONERS** | | | | | | | | | |
| OpenAI o1 (Jaech et al., 2024) | 52.2 | 91.0 | 91.0 | 77.0 | 60.0 | 98.0 | 88.0 | 82.1 | 79.9 |
| OpenAI o3-mini (Zhang et al., 2025a) | 40.3 | 94.0 | 93.0 | 74.0 | 55.0 | 96.3 | 100.0 | 77.6 | 78.8 |
| Claude-3.7-Sonnet (Anthropic, 2025) | 65.7 | 92.8 | 90.0 | 74.7 | 58.0 | 76.2 | 97.0 | 81.5 | 79.5 |
| DeepSeek-R1 (Guo et al., 2025) | 71.6 | 40.0 | 27.0 | 72.7 | 49.0 | 89.7 | 98.3 | 85.0 | 66.7 |
| **BEHAVIORAL CLONING** | | | | | | | | | |
| SFT w/o CoT (Wei et al., 2022a) | 37.5 | 96.0 | 88.8 | 73.4 | 74.8 | 37.5 | 85.9 | 76.1 | 71.2 |
| SFT w/ Short CoT (Yeo et al., 2025) | 77.6 | 92.6 | 87.0 | 82.9 | 73.8 | 54.8 | 87.6 | 88.0 | 80.9 |
| SFT w/ Long CoT (Yeo et al., 2025) | 83.5 | 95.6 | 89.2 | 83.4 | 76.6 | 68.6 | 79.6 | 79.1 | 81.9 |
| **ADVANCED RLVRs** | | | | | | | | | |
| GRPO (Shao et al., 2024) | 73.1 | 95.4 | 96.4 | 72.3 | 66.6 | 36.3 | 90.3 | 70.1 | 75.0 |
| Dr. GRPO (Liu et al., 2025b) | 68.6 | 96.0 | 95.6 | 73.9 | 75.4 | 32.1 | 84.3 | 65.6 | 73.9 |
| DAPO (Yu et al., 2025) | 86.5 | 96.0 | 94.8 | 80.9 | 65.8 | 40.0 | 95.3 | 74.6 | 79.2 |
| **CURRICULUM LEARNING** | | | | | | | | | |
| Data-balance RL (Parashar et al., 2025) | 86.5 | 95.8 | 95.6 | 76.8 | 64.4 | 45.6 | 95.3 | 73.1 | 79.1 |
| Easy-to-hard RL (Parashar et al., 2025) | 88.0 | 96.2 | 96.8 | 78.9 | 66.6 | 46.3 | 96.0 | 74.6 | 80.4 |
| ADARFT (Shi et al., 2025a) | 92.5 | 96.0 | 97.4 | 81.8 | 64.4 | 44.6 | 96.6 | 80.5 | 81.7 |
| **RULEREASONER (Ours)** | | | | | | | | | |
| RULEREASONER-4B | $82.0_{0.4}$ | $95.0_{0.6}$ | $96.3_{0.3}$ | $78.9_{0.8}$ | $66.6_{0.4}$ | $38.6_{0.5}$ | $96.3_{0.2}$ | $80.5_{0.7}$ | $79.2_{0.6}$ |
| RULEREASONER-8B | $\mathbf{95.5}_{0.3}$ | $\mathbf{96.4}_{0.4}$ | $\mathbf{97.0}_{0.2}$ | $\mathbf{84.7}_{0.6}$ | $70.4_{0.1}$ | $46.8_{0.3}$ | $98.3_{0.4}$ | $83.5_{0.3}$ | $\mathbf{84.0}_{0.5}$ |

8B surprisingly outperforms with a large performance gap. Notably, *on eight ID tasks*, OpenAI-o1 lags behind RULEREASONER-8B with $4.1\%$ point, whereas Claude-3.7-Sonnet underperforms with $4.5\%$ point. Also, RULEREASONER-8B outperforms prior strong RBRs such as HtT and Chain-of-Logic, which are built directly on top of OpenAI o3-mini for all tasks, except for AR-LSAT and Logical Deduction. This implies that **RULEREASONER benefits from RLVRs to obtain higher improvement in rule understanding and utilization**. In addition, RULEREASONER-8B also outperforms recent RLVR methods which are trained with lower intra-task performance variance in *eight tasks*, for instance, higher performance of 84.0% (+4.8%) yet with a lower variance of 3.1% (-0.5%), comparing to DAPO (79.2% with a variance of 3.6%). This demonstrates that **RULEREASONER not only develops impressive task performance, but also maintains the domain performance balance**.

**Out-of-Distribution Performance.** As illustrated in Figure 1, RULEREASONER-8B surpasses frontier LRMs across *three OOD benchmarks*. Specifically, it shows a remarkable 10.4% improvement compared to OpenAI-o1. As depicted in Table 3, RULEREASONER-8B consistently increases performance across the three OOD benchmarks, achieving the highest average performance gains of $\Delta56.0\%$, including $\Delta71.4\%$ on BBH, $\Delta48.4\%$ on ProverQA, and $\Delta48.2\%$ on BBEH. These findings highlight the effectiveness of RULEREASONER in enhancing the general rule-based reasoning capabilities of models. As shown in Table 3, the SFT baseline lags behind RULEREA-

Table 3: Comparison of average improvement. % denotes ID and % denotes OOD performance, respectively. Unlike the task-focused settings in §4.1, † indicates a full-set mix supervised training to obtain stronger OOD performance for SFT.

| Model | Pass@1 | Avg. $\Delta$ |
|---|---|---|
| Qwen3-8B | 27.4 / 34.2 | – |
| + SFT† | 81.9 / 66.6 | 54.5 / 34.4 |
| + GRPO | 75.0 / 75.8 | 47.6 / 41.6 |
| **+ Ours** | **84.0** / **81.7** | **56.6** / **47.5** |

SONER in both ID and OOD evaluations. Notably, while SFT improves ID performance to closely match RULEREASONER (81.9% versus 84.0%), its OOD performance remains significantly lower (34.4% versus 54.5%). We conclude that, in contrast to RLVR, SFT does not effectively generalize to unseen rules or OOD scenarios, which is also aligned to Chu et al. (2025). Extended evaluations on other reasoning tasks (AIME 2025, GPQA, and Coin Flip) and the test-time scaling are discussed in §C.6 and §C.3, respectively. Moreover, a relative OOD demonstration in Table 4 reveals that **RULEREASONER elicits models to extrapolate to new tasks by applying unseen rules, through a concise and logically rigorous reasoning trajectory.**

## 5 ANALYSES

### 5.1 ADVANTAGES OF DOMAIN-AWARE DYNAMIC SAMPLING

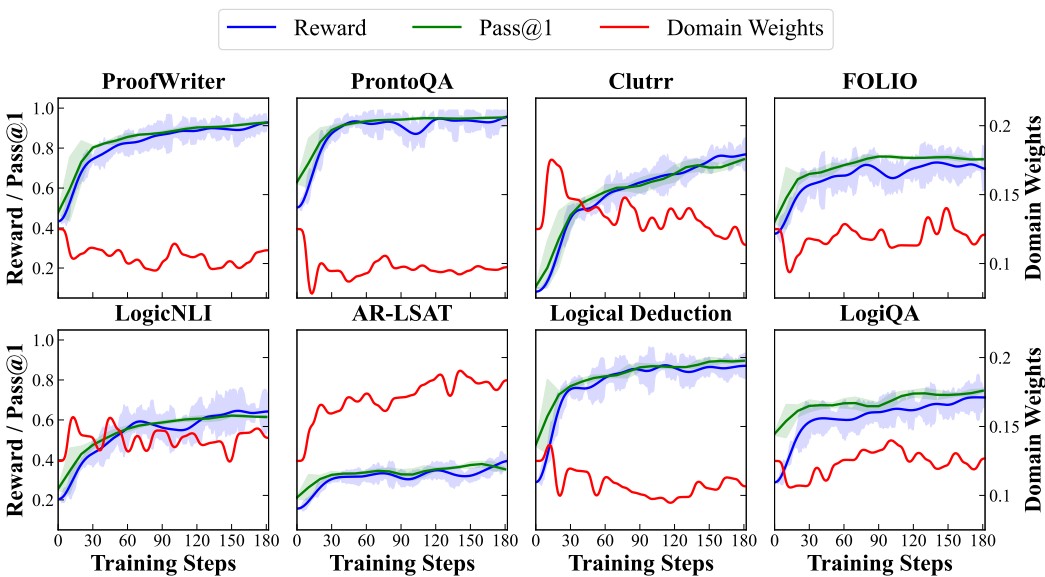

Figure 4: Learning dynamics by domains. "Reward" represents the training reward obtained from tasks and "Pass@1" denotes validation pass@1 performance. We employ exponential moving average smoothing for clearly displaying the curves "Reward", "Pass@1", and "Domain Weights".

As depicted in Figure 4, we find that RULEREASONER enhances task performance across all domains without tradeoff. Specifically, RULEREASONER takes care of the underperformed task such as AR-LSAT and increases its domain weights consistently. Even for the low-portion domains (*e.g.*, ProofWriter) and the decreased domains such as Clutrr, RULEREASONER still steadily improves their training rewards and validation pass@1 without reaching a plateau. Interestingly, analogous to the phenomenon described by Zucchet et al. (2025), the knowledge acquisition period during pre-training is accelerated fast on transition, but led to overfitting by the imbalanced data distributions. As shown in Table 6, we compare the OOD performance with DAPO and ADARFT to directly demonstrate that DADS achieves superior generalization without requiring external difficulty estimation (as in ADARFT) or additional rollout compute (as in DAPO).

Furthermore, we perform curriculum learning baselines as stated in §C.8. In summary, DADS surpasses these baselines by a significant margin (see Table 2 and 10). Static curriculum methods fail because they treat easy and hard domains equally or experience vast distribution shift during training, while DADS acts as an online scheduler, shifting compute resources from converged (easy) domains (*e.g.*, ProntoQA) to under-optimized (hard) domains (*e.g.*, AR-LSAT). This prevents over-optimization on solved tasks while preventing under-fitting on complex ones. In summary, we conclude that DADS serves as an online data scheduling strategy, stablizing the dynamics of on-policy RL training and mitigates over-optimization.

### 5.2 IMPACT OF TASK MIXING RECIPE

Figure 5 illustrates the impact of mixing recipes of incremental tasks in OOD tasks across models of different sizes. The incremental mixing strategies show consistent gains as the model size increases, indicating that a larger model capacity generally contributes to better generalization. The mixed collection of various reasoning types enhances the OOD performance nearly to 80% while the baseline "w/o All" at around 25%. It further validates the benefit of our training data curation principles and the effectiveness of task mixing for learning generalization.

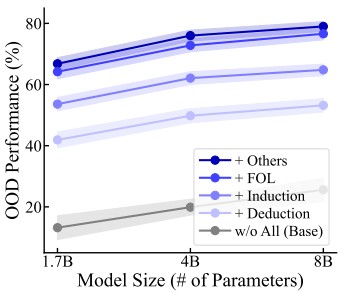

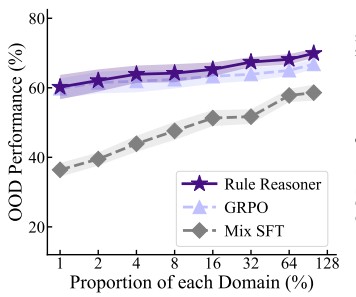

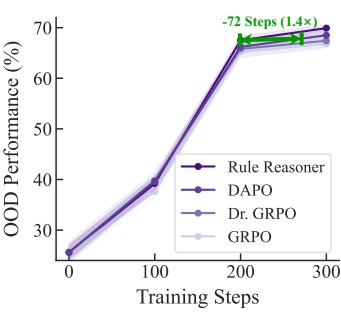

Figure 5: Impact on incremental task mixing recipes.

Figure 6: Impact on training sample efficiency.

Figure 7: Comparisons on different RLVR methods.

## 5.3 INVESTIGATION ON TRAINING SAMPLE EFFICIENCY

We assess OOD performance by scaling training sample sizes among three methods at step 500, sampling data with varying domain proportions on a log scale. It is evident that for all methods (Figure 6), as the proportion of each domain increases, RULEREASONER consistently outperforms GRPO and Mix SFT at every proportion level. For the same proportion of data, RULEREASONER achieves higher performance than Mix SFT as well as a slight advantage over GPRO. For example, when the domain proportion is 1%, RULEREASONER achieves an OOD performance of about 60% point, while the other two is around 59% point and 37% point. This implies the effectiveness of the online data scheduling stated in Section 3.

As shown in Figure 7, we also prove that RULEREASONER improves sample efficiency with fewer training steps than the existing RLVR methods. It achieves comparable OOD performance with ∼66.8% on average, roughly fewer 72 steps and approximately 1.4× acceleration compared to DAPO within the same training steps. Similarly, GRPO requires more steps to match RULEREASONER's performance, which indicates that the dynamic batch-level data curricula in RULEREASONER benefit RLVR and accelerate it to achieve higher performance with fewer costs.

## 5.4 CASE STUDY OF RULE APPLICATION

Table 4: Reasoning chains before and after training by RULEREASONER for comparison.

| Case Type | Input | Before Training | After Training |
|---|---|---|---|
| Explicit Rules (ProofWriter) | **Rules**: If something is smart then it is furry. If something is cold it is green. Cold, smart things are green. All furry things are white. If something is green, it is smart. All furry things are green. All cold, green things are rough. Cold things are green. **Facts**: Dave is blue. Dave is cold. Dave is green. **Question**: Dave is furry? | Step 1: Analyze the given rules and facts. Rule 1: If something is smart, then it is furry. ... Fact 1: Dave is blue. Fact 2: Dave is cold. ... Step 2: Determine if Dave is furry based on the given rules and facts. From Fact 2 and Rule 2, we know that Dave is green. From Fact 2 and Rule 8, we know that Dave is green. However, we also have Rule 4 stating that all furry things are white, and Dave is blue (Fact 1). This creates a contradiction, as Dave cannot be both furry and blue. Since we have a contradiction, we cannot determine if Dave is furry. The answer is Unknown. Answer: Unknown. ✗ | Step-by-Step Reasoning: 1. Dave is cold. From Rule 8: If something is cold, then it is green. Since Dave is cold, Dave is green. This is consistent with the given fact that Dave is green. From Rule 5: If something is green, then it is smart. Since Dave is green, Dave is smart. From Rule 1: If something is smart, then it is furry. Since Dave is smart, Dave is furry. From Rule 6: All furry things are green. This is consistent with the given fact that Dave is green. 3. Conclusion: Since Dave is smart (from Rule 5), and from Rule 1, if something is smart, then it is furry, we can conclude Dave is furry. Answer: True. ✓ |
| Implicit Rules (BBH) | **Context**: The following paragraphs each describe a set of three objects arranged in a fixed order. The statements are logically consistent within each paragraph. On a branch, there are three birds: a robin, an owl, and a hawk. The robin is to the right of the hawk. The owl is the leftmost. **Options**: (A) The robin is the rightmost (B) The owl is the rightmost (C) The hawk is the rightmost. **Question**: Which of the following options is true? | To answer this question, let's analyze the information provided in the context step by step: 1. There are three birds on a branch: ... Now, let's evaluate the options: (A) The robin is the rightmost: This option is not true because the robin is to the right of the hawk, but not necessarily the rightmost bird. (B) The owl is the rightmost: This option is not true because the owl is the leftmost bird, not the rightmost. Based on the information provided, none of the options (A, B, or C) are true. However, since we must choose one of the given options, we can conclude that the correct answer is: (B) Please note that this answer is not entirely accurate, as the owl is actually the leftmost bird. The question might have a mistake or missing information. Answer: B. ✗ | To determine the correct order of the three birds on the branch, let's analyze the information provided: 1. The owl is the leftmost. This means the owl is in the first position on the branch. 2. The robin is to the right of the hawk. This means the robin comes after the hawk in the order. Now, let's determine the positions of the hawk and the robin: Since the owl is the leftmost, the owl must be in the first position. The robin is to the right of the hawk, so the hawk must be in the second position, and the robin must be in the third position. So, the order from left to right is: 1. Owl 2. Hawk 3. Robin Now, let's evaluate the options: (A) The robin is the rightmost. So, this statement is true. (B) The owl is the rightmost. ... So, this statement is false. Answer: A. ✓ |

We show examples from RULEREASONER-8B in various test sets in Table 4 to demonstrate how our training recipe improves the performance of the model. Before training, the model fails to solve the problems because of various reasons, such as poor reasoning logic, reasoning distracted by the noisy rules, *etc*. After training, the model is able to derive a valid reasoning chain and answer rigidly. Additionally, we provide more successful and failed cases with their trajectories in Table 13 and 14.

### 5.5 GENERALIZATION MECHANISM FOR COMPOSITIONAL RULES

Motivated by the efficacy gap in Table 3, we analyze the origin of RULEREASONER's generalization capability. The OOD performance gains stem not just from the dynamic curriculum via DADS but also from the emergence of meta introspection during training, fostering self-simulation and self-verification abilities. RULEREASONER explores candidate paths and verifies logical consistency against inferred rules before concluding. This mirrors abstract introspection, where the model critiques intermediate outcomes and corrects flawed steps, enabling generalization beyond seen tasks.

In the first ProverQA example (Table 15), the untrained model makes unverified and speculative assumptions (*e.g.*, "that good intentions imply being lovable"). In contrast, the trained model adheres strictly to objective facts, noting that "lovable" is not explicitly defined or linked to those attributes in context. It analyzes available clues contextually, performing self-verification and consistency checks throughout the reasoning process. This rigorous behavior enables model to eliminate cognitive biases and logical fallacies, arriving at well-supported conclusions that align with established facts.

## 6 CONCLUSION

We introduce RULEREASONER, a training framework tailored to enhance both effectiveness and efficiency of multi-domain training for RLVR. It harmonizes diverse rule-based reasoning capabilities across various tasks, resulting in higher performance efficiently compared to existing rule-based reasoners, frontier LRMs, strong supervised baselines, and prior RLVR methods. We aim to explore further research into data-centric approaches to facilitate reasoning efficiency in future work.

**Limitations & Future Work.**   In this study, due to the scarce and imbalanced nature of rule-based reasoning data, current methods may not cover all rule formats and complexities found in real-world applications, which hinders task generalization. Besides, our method is constrained by the quality of rule filtering, particularly when dealing with noisy or redundant rules that can negatively impact reasoning. Furthermore, while effective with smaller models (4B and 8B), its scalability to large-scale modeling remains unverified due to computational limitations, despite potentially higher effectiveness in complex scenarios. Beyond these, the effectiveness and utility of our rule-applying in domain-specific scenarios remains to be explored and further enhanced. To meet the inherent nature of rule reasoning, the current methods presumably relies on a static or pre-defined set of rules. It lacks a mechanism for dynamic, ever-changing rule discovery and updating from new data or interactions, limiting its long-term adaptability in evolving environments. It could be studied in the future for moving towards a self-improving system. These limitations highlight areas for future improvement in expanding rule diversity and robustness to support longer reasoning trajectories.

## ETHICS STATEMENT

We adhere to ethical principles to ensure the responsible development and application of our proposed techniques. The research conducted in the article is in every respect in accordance with the ICLR code of ethics guidelines. Our work focusses on enhancing the rule-based reasoning abilities of models without directly involving human subjects or sensitive information. The study acknowledges ethical implications, such as the transparency of rule-based systems being advantageous for interpretability but raising concerns about accountability if rules are misapplied in high-stakes domains. We advocate for rigorous validation of rules against diverse datasets to avoid human biases in manually crafted rules. We also recognize the potential broader impacts, including the environmental and computational costs of LLM training, and strive to optimize our methods for efficiency.

## REPRODUCIBILITY STATEMENT

To ensure the reproducibility of our research, we provide detailed information regarding our methodology and experimental setup. The exact prompt templates utilized for datasets with both explicit and implicit rules during training and evaluation are detailed in Appendix A. Furthermore, Appendix B.1 provides a comprehensive list of the data sources for both training and OOD evaluation, along with specifics on how the datasets were curated for the RULECOLLECTION-32K. Our evaluation metrics are defined in §4.1, with additional evaluation results and analyses presented in Appendix C. Finally, a complete breakdown of the computational infrastructure and all hyperparameter assignments for both training and inference can be found in Appendix D. The code, model, and data will be made publicly available upon completion of the review process.

## ACKNOWLEDGMENTS

We would like to thank Zixia Jia, Hengli Li, and Xubo Qin from BIGAI for their valuable contributions in discussing the project, and Tong Wu from BIGAI for his initial trials, helpful discussions on RLVR training, and the infra-system operation and maintenance. We also thank the reviewers for their insightful suggestions to improve the manuscript. This work presented herein is supported by the National Natural Science Foundation of China (62376031).

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

# Appendix

TABLE OF CONTENTS

## A  PROMPTS

In this work, we use the same prompt template for each dataset for model training and evaluation.

### A.1  PROMPTS FOR DATASET WITH EXPLICIT RULES.

**Instruction:** Please answer the question based on the given rules and facts using either of [A/B/C/D] (or [True/False/Unknown]). Fill in the answer between <answer> and </answer>. Provide your step-by-step reasoning process between <think> and </think>.

**Input:**
- Rules: {{Rules}}
- Facts: {{Facts}}

**Question:** {{Question}}

**Options:** {{Options}} (OPTIONAL)

### A.2  PROMPTS FOR DATASET WITH IMPLICIT RULES.

**Instruction:** Please answer the question based on the given contexts using either of [A/B/C/D] (or [True/False/Unknown]). Fill in the answer between <answer> and </answer>. Provide your step-by-step reasoning process between <think> and </think>.

**Input:**
- Context: {{Context}}

**Question:** {{Question}}

**Options:** {{Options}} (OPTIONAL)

## B  DATA DETAILS

### B.1  DATA SOURCES

We list the training and evaluation data sources associated with the urls used in the paper as below. The followings are the training and validation data sources:

- ProofWriter (2021): https://allenai.org/data/proofwriter
- ProntoQA (2023): https://github.com/asaparov/prontoqa
- Clutrr (2019): https://github.com/SiyuanWangw/RuleApplication/blob/master/Data/clutrr
- AR-LSAT (2022): https://github.com/SiyuanWangw/RuleApplication/blob/master/Data
- FOLIO (2024): https://github.com/Yale-LILY/FOLIO/blob/main/data/v0.0
- LogicNLI (2021): https://github.com/omnilabNLP/LogicNLI/blob/main/dataset
- LogicalDeduction (2024): https://github.com/Aiden0526/SymbCoT/tree/main/data
- LogiQA (2023): https://github.com/csitfun/LogiQA2.0/blob/main/logiqa/DATA/LOGIQA

The followings are the OOD test data sources:

- BigBench-Hard (2023): https://huggingface.co/datasets/lukaemon/bbh
- ProverQA (2025): https://huggingface.co/datasets/opendatalab/ProverQA
- BigBench-Extra-Hard (2025): https://github.com/google-deepmind/bbeh

## B.2 DATASET CURATION DETAILS

Table 5: Data statistics of curated tasks. † denotes it can be deemed as deduction reasoning since we provide rules explicitly. The abbr. in the table indicate Modus Ponens (MP), Universal Instantiation (UI), Hypothetical Syllogism (HS), Disjunctive Syllogism (DS), Modus Tollens (MT), respectively. "FOL", "AR", "CS", and "CCR" denotes First-Order Logic, Analytical Reasoning, Constraint Satisfaction, and Categorical & Conjunctive Reasoning, respectively. "MC" represents multiple choice.

| Dataset | # Train/Test | Task Format | Reasoning Form | Reasoning Depth | Fiction Rule | Rule of Inference |
|---|---|---|---|---|---|---|
| ProofWriter (Tafjord et al., 2021) | 7,997/500 | Boolean | Deduction | [0, 5] | ✓ | MP, UI |
| ProntoQA (Saparov & He, 2023) | 8,000/500 | Boolean | Deduction | {1, 3, 5} | ✓ | UI, Conjunction Simplification |
| Clutrr (Sinha et al., 2019) | 268/67 | Free Text | Induction† | — | ✗ | HS |
| FOLIO (Han et al., 2024) | 1,208/242 | MC | FOL | [0, 7] | ✓ | MT, DS, UI |
| LogicNLI (Tian et al., 2021) | 8,000/500 | MC | FOL | [1, 5] | ✓ | MP, MT |
| AR-LSAT (Zhong et al., 2022) | 1,636/410 | MC | AR | — | ✓ | MP, MT |
| Logic. Dedu. (Xu et al., 2024a) | 1,200/300 | MC | CS | {1, 3, 5} | ✓ | MP, MT |
| LogiQA (Liu et al., 2023a) | 264/67 | MC | CCR | — | ✓ | MP, MT |

For ProntoQA, we randomly negate some of the proof questions to avoid learning the shortcut of answer "True". For ProofWriter, we randomly sample ten percent of the orginal source data considering the imbalance nature of the whole training data. Then we use DeepSeek-R1 to generate the reasoning process including short CoT and long CoT sequences for each QA sample. For LogiQA, we use data with the reasoning type both categorical reasoning and conjunctive reasoning that leverages the implicit rule application and reasoning. For BigBench-Hard, we use the subset of "logical_deduction" with three, five, and seven objects with varing levels of difficulties and select BoardgameQA and ZebraPuzzles from BigBench-Extra-Hard to keep consistent with our task definition for OOD evaluation.

## C ADDITIONAL EVALUATION RESULTS

### C.1 CHALLENGES OF DIFFERENT RULE SETTINGS

We investigate the task performance on Clutrr, with three-level rule settings of Figure 8 in the following: 1) Ordered Rules: rules are arranged in their application order; 2) Shuffled Rules: rules are provided in a random order; 3) Noisy Rules: rules are shuffled and include irrelevant ones. aligns with real-world scenarios, rules may contain distractors. To our expectation, the task with ordered rules achieves the best performance among them likely due to the logical sequence aiding in task execution. Shuffled Rules, while still contain only the relevant rules but in a random order, show a moderate performance drop. Noisy rules result in the most significant performance reduction with the added complexity of redundant rules as distractors, highlighting the negative effect on task performance.

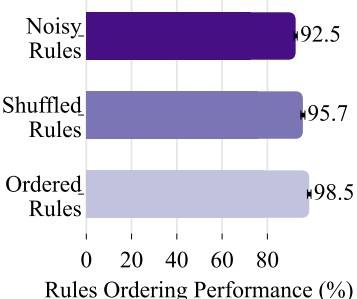

Figure 8: Comparison of performance on challenging rule settings.

## C.2 RULEREASONER CAN ADAPT TO VARYING RULE COMPLEXITY

As depicted in Figure 9, we present the extended OOD evaluation results, with test sets separated by rule complexity (*i.e.* task difficulty). The BBH, ProverQA, and BBEH benchmarks consist of questions requiring reasoning up to various difficulties that hinge on the diverse factors of query complexity.

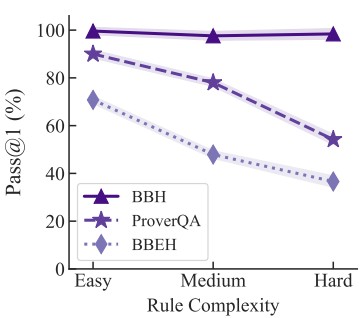

Specifically, for BBH, we divide the original test set into three difficulty levels based on the multi-hop number of the query. For ProverQA, we adopt the original difficulty levels from its source, which is separated by the number of reasoning steps. For BBEH, the test set was categorised into three levels according to the length of the query (in tokens): Easy [0, 1068), Medium [1068, 2175), and Hard [2175, 2741). Thus, we test the generalization capabilities on the three subsets of each OOD benchmark and report their performance on the higher difficulty questions. Not surprisingly, performance in easy

Figure 9: Comparison of performance on varying task complexity.

subset exhibit substantially stronger than corresponding medium- and hard-level subsets, with an average pass@1 of 86.7% compared to the 63.0% (-23.7%) of hard subset across benchmarks. Interestingly, we notice that RULEREASONER-4B drops significantly along with subsets in different difficulties, while we maintain the still performance in BBH. One possible explanation is that the base model, Qwen3-8B-Base, might have encountered a test set leakage, given that BBH was published in late 2022 but Qwen3 models were released in 2025 (Yang et al., 2025a). Therefore, we suggest assessing our models to more challenging benchmarks to achieve more reasonable results. We leave this direction for future work.

## C.3 TEST-TIME SCALABILITY

In the cutting-edge discussion on the essence and usefulness of a longer thinking process, Fatemi et al. (2025) find that the extra generated tokens do not help improve the final prediction accuracy, while Yeo et al. (2025); Yang et al. (2025c) hold the opposite positions which claim that accurate results are not necessary with the long reasoning process. To investigate this interesting question for rule-based reasoning, we study parallel test-time scaling strategies with RULEREASONER. We perform repeated sampling to investigate the upper limit of performance for each collection of rule-based reasoning problems in the way of Brown et al. (2024), illustrated in brown curves. Concretely, we take the majority vote and normalized weighted sum methods in the way of Wang et al. (2023).

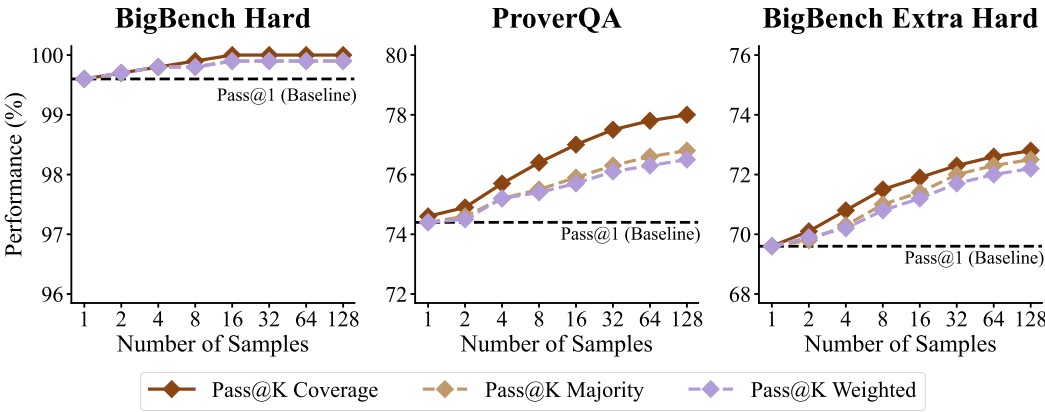

Figure 10: Comparison of OOD performance in parallel test-time scaling methods.

As depicted in Figure 10, test-time scaling demonstrates different effectiveness across benchmarks. For BigBench Hard, all Pass@K methods achieve near-perfect performance (close to 100%) with minimal scaling, indicating the limited complexity of the testbed to distinguish scaling benefits. In ProverQA, Pass@K Coverage consistently outperforms both the majority and weighted approaches,

with performance gaps widening as the sample size increases (~1.7% at 128 samples). BigBench Extra Hard reveals the most substantial scaling benefits, where the coverage method achieves 73.1% pass@k performance compared to ~72% for alternative approaches at 128 samples. The consistent superiority of Coverage sampling across challenging benchmarks (ProverQA and BigBench Extra Hard) suggests that diverse solution exploration outweighs consensus-based aggregation for complex reasoning tasks. These findings support the position that extended reasoning processes, when properly sampled, enhance the prediction accuracy of difficult problems in rule-based reasoning.

## C.4 COMPARISON WITH MORE BASELINES

We compare the differences and advantages of DADS versus DAPO and ADARFT on three OOD benchmarks. DAPO's dynamic sampling employs an online filtering scheme, which repeatedly samples and discards rollout generations until rewards fall within a target range, ensuring a mix of partially correct and incorrect answers. DADS is more efficient than DAPO with its dynamic sampling. DADS samples training problems before responses are generated, preventing wasted computation of creating and then discarding unwanted responses. ADARFT relies on an opaque deterministic difficulty estimation based on Qwen2.5-Math-7B (Yang et al., 2024), which limits the precision of initial difficulty judgement and introduces inductive bias from the LLM-as-a-judge model (Gu et al., 2024). For direct comparison, DADS only needs coarse-grained metadata labels from original data sources (it can be the form of "dataset-as-a-domain") for domain partitioning, reducing the need for prior dependencies like domain annotation or problem difficulty scoring.

| Model | BBH | ProverQA | BBEH | Average ($\Delta$) |
|---|---|---|---|---|
| Qwen3-8B-Base | 21.2 | 13.4 | 8.0 | 14.2 (-) |
| + DAPO (w/ dynamic sampling) | 95.4 | 68.8 | 62.0 | 75.4 (+61.2) |
| + ADARFT | 96.4 | 73.4 | 64.5 | 78.1 (+63.9) |
| + DADS (Ours) | **99.6** | **76.6** | **68.9** | **81.7 (+67.5)** |

Table 6: Performance comparison across different methods on BBH, ProverQA, and BBEH.

## C.5 ROBUSTNESS ANALYSES ON HYPERPARAMETERS OF DADS.

To investigate the robustness and sensitivity of the hyperparameters, we conduct concise analyses of the sensitivity and effect of the hyperparameters $\alpha$, $\tau$, and $\epsilon$ in DADS. Specifically, we keep two of the three hyperparameters constant and incrementally adjust the remaining hyperparameter. For each hyperparameter combination, we observe the model's performance on the BBH, ProverQA, and BBEH, three OOD benchmarks after convergence.

Table 7: Hyperparameter sensitivity analysis for different parameters

| Hyperparameter Sensitivity Analyses | | | | | |
|---|---|---|---|---|---|
| $\alpha$ | Avg. Pass@1 (OOD) | $\tau$ | Avg. Pass@1 (OOD) | $\epsilon$ | Avg. Pass@1 (OOD) |
| 0.1 | 68.6 | 0.1 | 67.6 | 0.1 | 70.4 |
| 0.3 | 69.3 | 0.3 | 69.4 | 0.3 | 70.2 |
| 0.5 | 70.4 | 0.5 | 70.1 | 0.5 | 69.9 |
| 0.8 | 70.1 | 0.8 | 70.4 | 0.8 | 69.3 |
| 1.0 | 69.8 | 1.0 | 69.7 | 1.0 | 68.9 |

As depicted in Table 7. We list the observations and effects as below.

- **Smoothing factor $\alpha$:** Performance shows an inverse "U-shape curve", initially increasing and then decreasing as $\alpha$ increases. Optimal performance is observed around $\alpha = 0.5$ with 70.4% pass@1. Larger $\alpha$ means more dependence on historical rewards. An excessively high or low $\alpha$ can degrade performance, suggesting a sweet spot for balancing historical and current rewards.

- **Temperature $\tau$ for magnitude control:** Performance generally improves as $\tau$ increases, with a slight dip at $\tau = 1.0$. The best performance is observed at $\tau = 0.8$ with 70.4% pass@1. Smaller $\tau$ indicates more sensitivity to reward fluctuations. Higher $\tau$ (up to a point) seem to improve performance, suggesting that less sensitivity to individual fluctuations can be beneficial.

- **Minimum sampling weight $\epsilon$:** Observation: Performance generally decreases as $\epsilon$ increases. The best performance is at $\epsilon = 0.1$ with 70.4% pass@1. Larger $\epsilon$ indicates a more average sampled number for each domain. This suggests that increasing the minimum sampling weight leads to a slight degradation in performance, implying that a lower $\epsilon$ (allowing for more varied sampling) is generally better.

## C.6 EXTRAPOLATING TO UNSEEN REASONING TASKS.

As presented in Table 8, we further explore the generalization to other verifiable domains, such as math, multi-disciplinary, and symbolic reasoning. We complement expanded inference-only OOD evaluation on more general reasoning tasks and show the extended OOD evaluation results including: (1) math (AIME 2025) (AIME, 2025); (2) multi-disciplinary (GPQA-Diamond) (Rein et al., 2024); and (3) symbolic (Coin Flip) (Wei et al., 2022b) reasoning benchmarks based on RULEREASONER-8B which is only trained on the rule-based reasoning tasks described in §3.3.

Table 8: Additional evaluation results on AIME 2025, GPQA (Diamond), and Coin Flip.

| Model | AIME 2025 | GPQA (Diamond) | Coin Flip | Average ($\Delta$) |
|---|---|---|---|---|
| Qwen3-8B-Base | 3.3 | 16.6 | 44.7 | 21.5 (-) |
| + DADS (Ours) | **23.3** | **44.9** | **95.1** | **54.4 (+32.9)** |

## C.7 EFFECTIVENESS OF DADS ON EXTREMELY HIGH DIFFICULTY.

To investigate whether DADS, the dynamic sampling approach, can still be a highly difficult learnable set of tasks by assigning a lot of computations to learn these samples, we identify the existence of an unlearnable set of extremely difficult tasks within our training domains, such as AR-LSAT. As Figure 4 illustrates, the training rewards for AR-LSAT start at approximately 0.11 and steadily increase to approximately 0.42. This improvement correlates with an increase in the training sample size, indicating a more assigned training computation.

Interestingly, even after Step 120, when the AR-LSAT domain was allocated nearly 20% of the computation in each training batch, other domains maintained their upward slope in reward/validation pass@1 curves. This leads us to conclude that the DADS algorithm achieves an optimal balance in computation assignment within multi-domain reinforcement learning dynamics. Although the experimental dashboard (Figure 4) might suggest an occasional over-emphasis on one domain, overall performance indicates an effective allocation strategy.

To empirically demonstrate it, we provide an additional comparison experiment between RULEREASONER with DADS and without DADS on the data mixing training (infusing the two-curve comparison of AR-LSAT).

We perform GRPO-style RLVR training with or without DADS on our curated rule-based reasoning dataset. We conduct interval evaluations every 50 training steps to assess the reasoning accuracy (Pass@1) on AR-LSAT, which we assume is an extremely hard domain to optimize with DADS. Finally, we compare the results of two setups and showcase the step-wise gains of validation performance. The comparison results between "w/o DADS" and "w/ DADS" assessed on AR-LSAT are shown in Table 9.

## C.8 COMPARISON WITH DATA SCHEDULING AND BALANCING BASELINES

We provide additional simple data scheduling baselines including Easy-to-hard SFT (Sun et al., 2024), Easy-to-hard RL (Parashar et al., 2025), and Data-balance RL (Parashar et al., 2025) for comprison purposes with DADS.

Table 9: Performance comparison of Pass@1 with and without DADS across different training steps.

| Training Steps | w/o DADS | w/ DADS (Δ) |
|---|---|---|
| 50 | 30.0 | 32.6 (+2.6) |
| 100 | 30.9 | 36.8 (+5.9) |
| 150 | 34.3 | 37.0 (+2.7) |
| 200 | 35.1 | 38.5 (+3.4) |
| 250 | 34.8 | 39.0 (+4.2) |
| 300 | 38.0 | 41.7 (+3.7) |

**Baselines setup.**   **Easy-to-hard SFT / RL:** We perform progressive curriculum training using SFT or GRPO-style RL on partitioned data subsets, respectively. First, we adopt LLM-as-a-judge with a Qwen3-8B-Base to obtain difficulty scores of problems. This allows us to partition the training data into eight domains based on difficulty. We then train a Qwen3-8B-Base model without DADS on this offline-partitioned data, progressively moving from easier to harder problems with the same portion of training epochs for each data partition. **Data-balance RL:** We perform GRPO-style policy gradient-based RL training using a balanced domain setup. First, based on the training data statistics in Table 5, we calculate the average samples per domain across all eight domains as 3,571 problem samples. Second, we apply repeated domain-balanced sampling to ensure 3,571 sampled problems for each domain. This involves down-sampling domains with more samples than the average and up-sampling those with fewer. Finally, we disable DADS and initiate RL training from scratch with this data-balanced setup.

**Results.**   As shown in Table 10, we demonstrate that training with DADS outperforms other three baselines of Easy-to-hard SFT / RL and Data-balance RL recipes both the three OOD rule-based reasoning benchmarks and the ODD general reasoning benchmarks, such as AIME 2025, GPQA (Diamond), and Coin Flip.

Table 10: Performance comparison across different benchmarks and training methods

| | BBH | ProverQA | BBEH | Average (Δ) |
|---|---|---|---|---|
| Qwen3-8B-Base | 21.2 | 13.4 | 8.0 | 14.2 (-) |
| + Easy-to-hard SFT | 89.2 | 31.8 | 17.7 | 46.2 (+32.0) |
| + Easy-to-hard RL | 96.0 | 72.2 | 63.5 | 77.2 (+63.0) |
| + Data-balance RL | 94.8 | 69.4 | 60.5 | 74.9 (+60.7) |
| + DADS (Ours) | **99.6** | **76.6** | **68.9** | **81.7 (+67.5)** |
| | **AIME 2025** | **GPQA (Diamond)** | **Coin Flip** | **Average (Δ)** |
| Qwen3-8B-Base | 3.3 | 16.6 | 44.7 | 21.5 (-) |
| + Easy-to-hard SFT | 6.6 | 33.8 | 48.7 | 29.7 (+8.2) |
| + Easy-to-hard RL | 20.0 | 39.8 | 91.4 | 50.4 (+28.9) |
| + Data-balance RL | 16.7 | 38.3 | 92.8 | 49.2 (+27.7) |
| + DADS (Ours) | **23.3** | **44.9** | **95.1** | **54.4 (+32.9)** |

## C.9   IMPACT OF RULE-BASED REASONING DATA CURATION

To the best of our knowledge, there is no accessible dataset used for training rule-based reasoners. We argue that current approaches over-specialize in math and code (Yu et al., 2025), omitting the scalability of general rule-based and natural language reasoning. This narrow data focus hinders progress on broader reasoning capabilities, a critical bottleneck we aim to address. Moreover, our collected multi-domain dataset, RULECOLLECTION-32K (§3.3), is integral to evaluate DADS, which specifically mitigates online training problem scheduling for such data in RLVR for LLMs.

Without these intrinsically linked multi-domain datasets, the full benefits of DADS can not be comprehensively demonstrated. As depicted in Table 11, additional experiments (RL on math-only data, RL on rule-only data, and RL jointly on both of them) using a naive GRPO objective (without DADS)

Table 11: Performance comparison on AIME 2025 and ProverQA across different training datasets.

| | AIME 2025 | ProverQA |
|---|---|---|
| Qwen3-8B-Base | 3.3 (-) | 13.4 (-) |
| + AIME 1983-2024 | 63.3 (+60.0) | 38.8 (+25.4) |
| + RULECOLLECTION-32K (Ours) | 23.3 (+20.0) | 73.6 (+60.2) |
| + AIME 1983-2024 + RULECOLLECTION-32K (Ours) | **66.7 (+63.4)** | **80.4 (+67.0)** |

further underscore the irreplaceable necessity of our curated data, proving it can not be substituted by training solely on existing math problems such as AIME 1983-2024 (Veeraboina, 2023).

## D COMPUTATIONAL ENVIRONMENTS

### D.1 INFRASTRUCTURE & HYPERPARAMETERS

We list the details of the computational environments of training and inference in Table 12.

Table 12: Hyper-parameters of RULEREASONER-4B and RULEREASONER-8B on-policy RL training and inference.

| Computational Infrastructure 4 × A100-SXM4-80GB GPU | | | |
|---|---|---|---|
| **Hyperparameter** | **Assignment** | **Hyperparameter** | **Assignment** |
| Base model | RULEREASONER-4B (Qwen3-4B-Base) | Base model | RULEREASONER-8B (Qwen3-8B-Base) |
| Training strategy | FSDP (Zhao et al., 2023) | Training strategy | FSDP (Zhao et al., 2023) |
| Training epochs | 3 | Training epochs | 3 |
| Training precision | bfloat16 | Training precision | bfloat16 |
| Mini batch size | 65,536 tokens | Mini batch size | 65,536 tokens |
| Micro batch size | 16,384 tokens | Micro batch size | 16,384 tokens |
| Max input length | 1,024 | Max input length | 1,024 |
| Max output length | 2,048 | Max output length | 2,048 |
| Max learning rate | $1e-6$ | Max learning rate | $1e-6$ |
| Optimizer | AdamW | Optimizer | AdamW |
| Adam beta weights | $0.9, 0.95$ | Adam beta weights | $0.9, 0.95$ |
| KL loss coefficient | 0 | KL loss coefficient | 0 |
| Rollout per sample | 64 | Rollout per sample | 64 |
| Clip ratio | 0.2 | Clip ratio | 0.2 |
| Gradient clipping | 1.0 | Gradient clipping | 1.0 |
| Evaluation steps | 10 | Evaluation steps | 10 |
| Save steps | 10 | Save steps | 10 |
| Training engine | PyTorch Paszke et al. (2019) | Training engine | PyTorch (Paszke et al., 2019) |
| Decay factor $\alpha$ | 0.5 | Decay factor $\alpha$ | 0.5 |
| Base reward $\epsilon$ | 0.1 | Base reward $\epsilon$ | 0.1 |
| Scaling factor $\tau$ | 0.8 | Scaling factor $\tau$ | 0.5 |
| Rollout engine | vLLM (Kwon et al., 2023) | Rollout engine | vLLM (Kwon et al., 2023) |
| Rollout temperature | 0.8 | Rollout temperature | 0.8 |
| Rollout top-p | 1 | Rollout top-p | 1 |
| Random seeds | $\{21, 42, 84\}$ | Random seeds | $\{21, 42, 84\}$ |

## E EXAMPLE TRAJECTORIES

We provide additional positive and negative examples (*i.e.*, failed cases) with fine-grained trajectories from various tasks to demonstrate clear step-by-step logical reasoning patterns and well-organized thought processes of RULEREASONER after training in Table 13, Table 14, and Table 15.

## E.1 IN-DISTRIBUTION POSITIVE EXAMPLES

Table 13: In-distribution positive examples of RULEREASONER on various datasets.

| Dataset | Input | RULEREASONER |
|---|---|---|
| Clutrr | **Context**: James took his son Francisco out on a hunting trip. Natasha, Francisco's daughter, wanted to come, but they thought she was too young. Instead, James's wife Frances stayed home with her.
**Question**: How is Natasha related to Frances? | To determine how Natasha is related to Frances, let's analyze the information provided and trace the family relationships step by step. Given Information:1. James took his son Francisco out on a hunting trip. This means James is Francisco's father ... Step-by-Step Reasoning: 1. James is Francisco's father, Francisco is James's son. 2. Since Frances is James's wife, she is also Francisco's mother. 3. Since Francisco is James's son and Frances's son, Natasha is Francisco's daughter, making Natasha James's granddaughter and Frances's granddaughter.4. Conclusion: Natasha is the granddaughter of Frances.
Answer: granddaughter ✓ |
| Folio | **Context**: Diamond Mine is a professional wrestling stable, formed in WWE. Roderick Strong leads Diamond Mine. Diamond Mine includes the Creed Brothers, and Ivy Nile. Imperium has a feud with Diamond Mine.
**Question**: Roderick strong leads a professional wrestling stable. | To determine whether the statement "Roderick Strong leads a professional wrestling stable" is true, false, or unknown based on the given context, let's analyze the information provided: 1. The context states: ... 3. From the context, we know that Roderick Strong leads Diamond Mine, and Diamond Mine is a professional wrestling stable. 4. Therefore, since Roderick Strong leads Diamond Mine, and Diamond Mine is a professional wrestling stable, it follows that Roderick Strong leads a professional wrestling stable.
Answer: true ✓ |
| Logical Deduction | **Context**: A fruit stand sells five fruits: apples, loquats, cantaloupes, watermelons, and kiwis. The watermelons are more expensive than the loquats. The watermelons are less expensive than the kiwis. The loquats are more expensive than the apples. The cantaloupes are the cheapest.
**Options**: A) The apples are the third-most expensive. B) The loquats are the third-most expensive. C) The cantaloupes are the third-most expensive. D) The watermelons are the third-most expensive. E) The kiwis are the third-most expensive.
**Question**: Which of the following is true? | To determine the correct order of the fruits from most expensive to least expensive, let's analyze the information provided: 1. The cantaloupes are the cheapest. This means cantaloupes are the least expensive. 2. The loquats are more expensive than the apples. Loquats ¿ Apples 3. The watermelons are more expensive than the loquats. Watermelons ¿ Loquats 4. The watermelons are less expensive than the kiwis. Kiwis ¿ Watermelons Now, let's piece this information together to establish the order from most expensive to least expensive: the order from most expensive to least expensive is: Kiwis ¿ Watermelons ¿ Loquats ¿ Apples ¿ Cantaloupes Now, let's evaluate the options: A) The apples are the third-most expensive. Apples are fourth in the order, so this is false. B) The loquats are the third-most expensive. Loquats are third in the order, so this is true. C) The cantaloupes are the third-most expensive. Cantaloupes are fifth in the order, so this is false. D) The watermelons are the third-most expensive. Watermelons are second in the order, so this is false. E) The kiwis are the third-most expensive. Kiwis are first in the order, so this is false
Answer: B ✓ |
| ProverQA | **Context**: Every protozoa is either parasitic or free-living, but not both. Frederick lives in a host. Any protozoa that lives in a host or feeds on nutrients is parasitic. Queenie lives in a host. Queenie does not feed on nutrients.
**Options**: A) True B) False C) Uncertain
**Question**: Queenie is free-living. | To determine whether the statement "Queenie is free-living", let's analyze the information provided and the options step by step. Given Information: 1. Every protozoa is either parasitic or free-living, but not both. This means if a protozoa is parasitic, it cannot be free-living, and vice versa ... Step-by-Step Reasoning: 1. Queenie lives in a host. From the given information, any protozoa that lives in a host or feeds on nutrients is parasitic. Since Queenie lives in a host, she falls into the category of protozoa that lives in a host. 2. Any protozoa that lives in a host or feeds on nutrients is parasitic. Since Queenie lives in a host, she is parasitic. 3. Every protozoa is either parasitic or free-living, but not both. If Queenie is parasitic, she cannot be free-living. 4. Conclusion: Since Queenie is parasitic, she is not free-living.
Answer: B ✓ |
| LogiQA | **Context**: Customs is a state organ that supervises, checks and collects customs duties on all goods and articles entering and leaving the country.
**Options**: A) Check other people's luggage B) Check the body of others C) Take portraits of others D)Detain others for one day
**Question**: According to the above definition, when on duty, customs officer can? | To determine the correct answer, we need to analyze the role of a customs officer based on the given context and see which option aligns with their duties. Reasoning: 1. The primary role of a customs officer is to supervise, check, and collect customs duties on goods and articles entering and leaving the country. 2. To ensure compliance with customs regulations, customs officers may need to inspect luggage or packages to verify the contents and ensure they are properly declared. 3. Checking the body of others (Option B) is not a standard duty of a customs officer, as it would involve invasive procedures that are not typically part of customs inspection. 4. Taking portraits of others (Option C) is unrelated to the duties of a customs officer. 5. Detaining others for one day (Option D) is a more severe action and would require specific legal grounds, which are not mentioned in the context of customs duties. Conclusion: The most appropriate action for a customs officer, based on their role, is to check other people's luggage (Option A) to ensure compliance with customs regulations.
Answer: A ✓ |
| ProntoQA | **Rules**: Every vumpus is a rompus and a sterpus. Every rompus is a brimpus and a lempus. Zumpuses are wumpuses and dumpuses. Each grimpus is a shumpus and a tumpus. Each lorpus is muffled and an impus. Every numpus is a gorpus. Gorpuses are zumpuses and jompuses. Every jompus is a vumpus and a grimpus.
**Facts**: Alex is a brimpus and a lempus. Alex is a zumpus and a gorpus
**Question**: Alex is not a vumpus? | We need to analyze the information provided and trace Alex's properties through the relationships defined. Given Rules: 1. Every vumpus is a rompus and a sterpus. 2. Every rompus is a brimpus and a lempus ... Given Facts: Alex is a brimpus and a lempus. Alex is a zumpus and a gorpus. Step-by-Step Reasoning: 1. Alex is a zumpus and a gorpus. From rule 7: Every gorpus is a zumpus and a jompus. Since Alex is a gorpus, Alex is a zumpus and a jompus. From rule 3: Zumpuses are wumpuses and dumpuses. Since Alex is a zumpus, Alex is a wumpus and a dumpus. From rule 8: Every jompus is a vumpus and a grimpus. Since Alex is a jompus, Alex is a vumpus and a grimpus. 2. Alex is a vumpus. From the above reasoning, since Alex is a jompus, and every jompus is a vumpus, Alex must be a vumpus. 3. Conclusion: The statement "Alex is not a vumpus" would be false because we have determined that Alex is indeed a vumpus.
Answer: false ✓ |

## E.2 IN-DISTRIBUTION NEGATIVE EXAMPLES

Table 14: In-distribution negative examples of RULEREASONER on various datasets.

| Dataset | Input | RULEREASONER |
|---|---|---|
| LogiQA | **Context**: As far as the role of individual and nation is concerned, cultural quality mainly infiltrates people's mind and improves people's realm by forming a specific cultural atmosphere; In other words, the role of cultural quality is indirect and abstract **Options**: A)Cultural quality is more important than humanistic spirit B)Humanistic spirit is more important than cultural quality C)Cultural quality and humanistic spirit are one thing D)When vigorously improving the quality of national culture, we should attach great importance to the construction of humanistic spirit **Question**: The function of humanistic spirit is more direct and specific | Context Analysis: The role of cultural quality is described as indirect and abstract. It mainly infiltrates people's minds and improves their realm by forming a specific cultural atmosphere. The function of humanistic spirit is described as more direct and specific. Question Analysis: The question asks us to compare the importance of cultural quality and humanistic spirit based on their functions. It suggests that since the humanistic spirit is more direct and specific, we should consider its importance in relation to cultural quality. Reasoning: The context tells us that cultural quality is indirect and abstract, while the humanistic spirit is direct and specific. Since the humanistic spirit is more direct and specific, it likely has a more immediate and tangible impact compared to cultural quality, which is more indirect and abstract. Therefore, it would be reasonable to conclude that the humanistic spirit is more important than cultural quality in terms of its direct and specific function. Conclusion: Given that the humanistic spirit is more direct and specific, it is more important than cultural quality. Answer: B ✗ |
| AR-LSAT | **Context**: While on vacation, Sukanya receives several e-mail messages from work, each message from one of three associates: Hilary, Jerome, and Lula. Sukanya receives at least one and no more than two messages from each of them. Sukanya receives each message on the day it is sent. No more than one message is sent each day. The messages are received in a manner consistent with the following: The first message is not from Lula. Both the first and last messages are from the same person. Exactly once Sukanya receives a message from Jerome on the day after receiving one from Hilary. Of the first three messages, exactly one is from Jerome. **Options**: A) Exactly one of the messages is from Lula. B) Exactly two of the messages are from Jerome. C) The second message is from Lula. D) The third message is from Hilary. E) The fourth message is from Jerome. **Question**: If Sukanya receives exactly four e-mail messages, then which one of the following must be true? | Step-by-Step Reasoning: 1. The first message is not from Lula. Both the first and last messages are from the same person Therefore, the last message must also be from the same person as the first message. So, the first and fourth messages are from either Hilary or Jerome. 2. Messages from Jerome: Since there is only one Jerome message in the first three messages, the Jerome message must be in one of the first three messages. The Jerome message on the day after Hilary must be the fourth message (since there are only four messages). 3. Messages from Lula: Sukanya receives at least one and no more than two messages from each associate. Since the first message is not from Lula, Lula must be in the second, third, or fourth message. If the first and fourth messages are from Hilary, then the second and third messages must include Lula and Jerome. If the first and fourth messages are from Jerome, then the second and third messages must include Hilary and Lula. 4. Analyzing the Options: Option A: Since Sukanya receives at least one message from each associate and there are four messages, it's possible that Lula sends two messages. This option is not necessarily true. Option B: The first and fourth messages are from the same person (either Hilary or Jerome). If the first and fourth messages are from Jerome, then Jerome sends two messages. If the first and fourth messages are from Hilary, then Jerome must send one message in the first three messages and one in the fourth message. Therefore, Jerome could send either one or two messages. This option is not necessarily true. Option C: The second message could be from Lula, but it's not necessarily true based on the constraints. This option is not necessarily true. Option D: The third message could be from Hilary, but it's not necessarily true based on the constraints. Option E: Since exactly once Sukanya receives a message from Jerome on the day after receiving one from Hilary, and there are only four messages, the Jerome message on the day after Hilary must be the fourth message. Therefore, the fourth message must be from Jerome. Answer: E ✗ |
| LogicNLI | **Rules**: If there is at least one people who is popular, then Curt is not unable. If there is someone who is loyal, then Brock is timid and Lucille is not popular. If there is someone who is timid, then Marcus is unable and Lucille is loyal. Garth being not shiny is equivalent to Jed being not unable and Marcus being light. If someone who is not timid is also loyal, then he is not unable. If there is at least one people who is light or not shiny, then Silas is timid. If Lucille is light, then Brock is not popular. If Curt is not light, then Marcus is popular. Silas is light if and only if Silas is loyal and Brock is not unable. If someone is not timid, then he is loyal. If there is someone who is both unable and not timid, then Jed is shiny. Someone who is both not shiny and unable is always light. **Facts**: Silas is light. Marcus is not timid. Marcus is popular. Marcus is light. Lucille is not unable. Curt is loyal. Brock is unable. Jed is loyal. Curt is unable. Garth is light. Brock is not shiny. Jed is popular. **Question**: Brock is unable. | Given Facts: 1. Silas is light. 2. Marcus is not timid. ... Given Rules: 1. If there is at least one person who is popular, then Curt is not unable. 2. If there is someone who is loyal, then Brock is timid and Lucille is not popular. ... Analysis: From the given facts, we can see that **Fact 7** directly states: Brock is unable. This means that the statement "Brock is unable" is **entailed** by the facts provided. Conclusion: Since the fact "Brock is unable" is explicitly given in the facts, the statement "Brock is unable" is **entailed** by the information. Answer: entailment ✗ |

## E.3 OUT-OF-DISTRIBUTION EXAMPLES

Table 15: OOD trajectories before and after training by RULEREASONER for comparison.

| Dataset | Input | Before Training | After Training |
|---|---|---|---|
| ProverQA | **Context**: Colten is coordinated. If Colten has good intentions, then he is either good-natured or well-meaning (but not both). Colten is either coordinated or clumsy, but not both. Colten is good-natured. Colten has good intentions. **Options**: A) True B) False C) Uncertain **Question**: Colten is lovable? | Step: 1. This tells us that Colten is not clumsy, as he is either coordinated or clumsy, but not both. 2. This is a conditional statement. If the condition (good intentions) is true, then one of the two outcomes (good-natured or well-meaning) must be true, but not both. 3. This is a tautology based on the first statement, so it doesn't provide new information. 4. This directly tells us that Colten is good-natured. 5. This is the condition in the second statement, so we can apply it. We know from the fourth statement that Colten is good-natured. ... The term lovable is subjective and not directly defined in the given context. However, based on the information provided, we can infer that Colten is a good person with good intentions and a good nature. These traits generally contribute to someone being considered lovable. Answer: A. ✗ | To determine whether the statement Colten is lovable is true, false, or uncertain based on the given context, let's analyze the information step by step. From statement 1: ... From statement 3: Colten is either coordinated or clumsy, but not both. 3.From statement 5: Colten has good intentions.4. From statement 2: If Colten has good intentions, then he is either good-natured or well-meaning (but not both). 5. From statement 4: Colten is good-natured. Therefore, from statement 2, since Colten is good-natured, he cannot be well-meaning (because it's either good-natured or well-meaning, but not both).We know that Colten is good-natured and has good intentions. However, the context does not provide any direct information about whether Colten is lovable. The attributes of being coordinated, good-natured, and having good intentions do not necessarily imply that Colten is lovable, as lovable is not explicitly defined or linked to these attributes in the given context. Therefore, based on the provided information, we cannot definitively determine whether Colten is lovable. Answer: C. ✓ |
| ProverQA | **Context**: Tommy has fine fleece. For all sheep, if a sheep has fine fleece, then it is either prized by farmers or produces quality wool (or both). Nolan has fine fleece. Tommy is not prized by farmers.Question: Based on the above information, is the following statement true, false, or uncertain? Tommy produces quality wool. **Options**: A) True B) False C) Uncertain **Question**: Tommy produces quality wool? | The answer is C) Uncertain. Step 1: Analyze the given information Tommy has fine fleece.For all sheep, if a sheep has fine fleece, then it is either prized by farmers or produces quality wool (or both). Nolan has fine fleece. Tommy is not prized by farmers.Step 2: Apply the given information to the question We know that Tommy has fine fleece, and according to the given information, if a sheep has fine fleece, it is either prized by farmers or produces quality wool (or both). We also know that Tommy is not prized by farmers.Step 3: Determine the answer Since Tommy is not prized by farmers, we cannot determine if he produces quality wool or not based on the given information. Answer: Uncertain. ✗ | To determine whether the statement Tommy produces quality wool is true, false, or uncertain based on the given context, let's analyze the information step by step. Reasoning:1. **From statement 1:** Tommy has fine fleece.2. **From statement 2 (general rule):** If a sheep has fine fleece, then it is either prized by farmers or produces quality wool (or both). Since Tommy has fine fleece (from statement 1), we can apply this rule to Tommy: Tommy is either prized by farmers or produces quality wool (or both).3. **From statement 4:** Tommy is not prized by farmers. This means the prized by farmers part of the rule is false for Tommy.4. **Applying the rule to Tommy:** Since Tommy is not prized by farmers, the only remaining possibility is that Tommy produces quality wool. Answer: A. ✓ |
| BigBench Hard | **Context**: The following paragraphs each describe a set of three objects arranged in a fixed order. The statements are logically consistent within each paragraph. On a shelf, there are three books: a yellow book, a brown book, and a red book. The red book is to the left of the brown book. The yellow book is to the right of the brown book. **Options**: (A) The yellow book is the rightmost (B) The brown book is the rightmost (C) The red book is the rightmost **Question**: Which of the following options is true? | The answer is C. Since the red book is to the left of the brown book, we can place the red book first and the brown book second. The yellow book is to the right of the brown book, so it must be placed third. Now, let's evaluate the options: (A) The yellow book is the rightmost: This is true, as the yellow book is placed third in the arrangement.(B) The brown book is the rightmost: This is false, as the brown book is placed second in the arrangement. (C) The red book is the rightmost: This is false, as the red book is placed first in the arrangement. Answer: C. ✗ | To determine which of the options is true, let's analyze the given context step by step. Step 1: Understand the relationships between the books. From statement 2: The red book is to the left of the brown book. This means: Red book ¡ Brown book From statement 3: The yellow book is to the right of the brown book. This means: Brown book ¡ Yellow book Step 2: Arrange the books in order. Combining the two relationships: Red book ¡ Brown book ¡ Yellow book So, the order from left to right is: Red book, Brown book, Yellow book. Step 3: Determine the rightmost book. In the order Red book, Brown book, Yellow book, the rightmost book is the Yellow book. Step 4: Evaluate the options. (A) The yellow book is the rightmost From our arrangement, the yellow book is indeed the rightmost. The red book is the leftmost, not the rightmost. This option is **false**. Answer: A. ✓ |

