# OpenReview forum: "RuleReasoner: Reinforced Rule-based Reasoning via Domain-aware Dynamic Sampling"
_ICLR.cc/2026/Conference — ICLR 2026 Poster_

### Official Review · Reviewer_VMSU · 2025-10-24

**Soundness:** 3
**Presentation:** 3
**Contribution:** 2
**Rating:** 4
**Confidence:** 4

**Summary:**

This paper introduces RuleReasoner, a reinforcement learning framework designed to enhance large language models’ rule-based reasoning ability. The authors construct a new dataset, RuleCollection-32K, covering eight logic and reasoning domains (e.g., ProofWriter, ProntoQA, AR-LSAT, LogiQA). The model is trained with Reinforcement Learning with Verifiable Rewards (RLVR) and a new sampling strategy called DADS (Domain-aware Dynamic Sampling), which adjusts training focus dynamically based on domain performance. Experiments on both in-distribution and out-of-distribution benchmarks show consistent improvements over baselines such as GRPO, DAPO, and even OpenAI o1-mini on some tasks.

**Strengths:**

The Domain-aware Dynamic Sampling methods is simple and experiments show that the effectiveness on enhancing the logical reasoning ability with RL.

**Weaknesses:**

The Domain-aware Dynamic Sampling also is not really something new. There are many works related to sampling data regarding to the weights of domains. It'd be more comvincing for the authors to discuss related works or the technical contribution of the paper is not enough.

**Questions:**

See Questions

---

> ### Author Response · Authors · 2025-11-20
>
> We sincerely appreciate your acknowledgement of simplicity and effectiveness of our method.
>
> **W1:** We discuss the related work to distinguish our contributions in the following.
> - Traditional curriculum learning relies on static data sampling with fixed schedules (e.g., easy-to-hard) or pre-defined heuristics (e.g., difficulty and task frequency). Recent works like DAPO [1] improves dynamic sampling by decoupling policy and value updates, enabling more stable exploration while it still generates large recompute cost in the rollout stage to filter out samples of 0/1 accuracy. Moreover, AdaRFT [2] explores an efficient batch-level sampling method using curriculum learning; however, it relies on human priors or an empirical success rate by models on sample difficulty.
> - In comparison, our Domain-aware Dynamic Sampling (DADS) fundamentally differs from existing approaches in the following aspects:
>   - **Contextual Rule Shuffling (§3.2)** dynamically alters the rules or constraints of the environment during training preventing overfitting to a fixed set of conditions.
>   - **Removing the KL Constraint (§3.2)** allows the policy to deviate more freely, enabling faster adaptation and domain randomization.
>   - Meanwhile, we **remove entropy bonus (§3.2)** to avoid extreamly high entropy to mitigate the issue of entropy explosion occured in the training procedure of RuleReasoner.
>   - Beyond these above-mentioned methods, **DADS (§3.1)** enables autonomous adaptation through reward feedback and innovates by dynamic curriculum with real-time adjustment of sampling weights via online policy gradients, responding to model’s evolving competency.
>
> Beyond DADS, our paper introduces:
> - A large and **novel dataset RuleCollection-32K (§3.3)**,  which spans diverse rule-centric reasoning tasks varying in format, reasoning forms, and complexity, allowing training and ID/OOD evaluation.
> - First to unify dynamic sampling into a **cohesive framework for rule/logical reasoning** demonstrating the framework’s effectiveness. Our experiments provide empirical evidence that our model outperforms frontier LRMs by a significant margin across both ID and OOD reasoning tasks with higher generalization and computational efficiency.
>
> **We expanded the section of related work (§2.1) to explicitly contrast DADS with existing domain-weighted methods, highlighting these innovations.**
>
> Thank you once again for your invaluable feedback!
>
> [1] Yu et al. DAPO: An Open-Source LLM Reinforcement Learning System at Scale. NeurIPS 2025.
>
> [2] Shi et al. Efficient Reinforcement Finetuning via Adaptive Curriculum Learning. NeurIPS 2025 Workshop on Efficient Reasoning. 2025.

---

> ### Comment · Reviewer_VMSU · 2025-11-25
>
> Thanks for the responses. They make sense to me. I have raised my score. Good luck!

---

> > ### Author Response · Authors · 2025-11-26
> >
> > We are glad our responses addressed your concerns.
> >
> > Thank you for your thoughtful questions and insightful discussion, which we will include in the final version.
> >
> > We appreciate your efforts throughout this process!

---

### Official Review · Reviewer_fFpE · 2025-10-28

**Soundness:** 3
**Presentation:** 3
**Contribution:** 3
**Rating:** 8
**Confidence:** 2

**Summary:**

The paper targets rule-based reasoning—an area where general language models especially struggle—and proposes RuleReasoner: a reinforcement learning with verifiable rewards (RLVR) framework augmented by Domain-aware Dynamic Sampling (DADS). DADS adaptively shifts training probability toward domains with lower historical rewards, balancing progress across tasks (domains) and mitigating over-specialization. The dataset spans eight domains with varied rule types and depths; training further removes KL loss to encourage diversity and shuffles rule order to discourage memorization. Experiments on BBH, ProverQA, and related suites show that RuleReasoner-8B surpasses strong baselines (including several frontier LLMs on selected tasks), while RuleReasoner-4B approaches or exceeds peer models at similar scale—suggesting practical sample/compute efficiency and SLM-friendliness.

**Strengths:**

* **Addresses a real challenge:** Domain imbalance in multi-domain RL training is a genuine problem. DADS provides an intuitive solution using historical reward signals to guide sampling without requiring human-designed curricula or expensive recomputation.

* **Strong OOD generalization:** The out-of-distribution results (Δ 10.4% over OpenAI-o1 on average across BBH, ProverQA, BBEH) demonstrate generalization beyond the training data. Consistency across multiple OOD benchmarks and additional tasks (AIME, GPQA, Coin Flip) strengthens this claim.

* **Comprehensive evaluation:** The paper includes appropriate baselines across multiple categories (frontier models, prior rule-based reasoners, SFT variants, RLVR methods), statistical testing with multiple seeds, ablation studies, and insightful learning dynamics analysis (Figures 4–6).

* **Reproducibility:** Detailed hyperparameters, data sources, and prompts are provided. The authors commit to releasing code and data.

**Weaknesses:**

1. **Missing competitive baseline AdaRFT:** The main results table (Table 2) includes many baselines from different viewpoints, but it omits AdaRFT and DAPO (with dynamic sampling), which is arguably the most competitive related work. AdaRFT (Adaptive Curriculum Reinforcement Finetuning) also uses a dynamic sampling strategy to balance training difficulty [1], making it directly comparable to the proposed approach. In the paper, AdaRFT appears only in Appendix Table 6 (evaluated on three OOD datasets) instead of the comprehensive results in Table 2.

2. **Unclear OOD generalization mechanism:** The method (RuleReasoner with DADS) demonstrates improved out-of-distribution performance (a 10.4% gain over three OOD benchmarks), but the reason for this improvement remains unclear. The authors attribute the ID gains to balancing training across domains (preventing any single task from dominating; Line 344) while attributing the OOD gains to extrapolation capability (Line 364), but this explanation is superficial and doesn’t reveal what the model actually learns. It’s uncertain whether the model is acquiring more general reasoning patterns or simply benefiting from a data augmentation effect (i.e., seeing more diverse examples). No qualitative analysis or insight into the model’s behavior on OOD inputs is provided beyond the aggregate accuracy numbers. This lack of explanation is problematic because robust generalization is a central claim of the paper. Prior work has shown that even advanced LLMs have significant gaps in handling complex or compositional rules [2], so the reader expects a deeper discussion of how this approach addresses those gaps.

3. **Insufficient explanation for DADS’s effectiveness:** The paper provides empirical evidence that it improves performance—e.g., Table 9 shows higher accuracy at various training steps with DADS compared to without DADS. However, the analysis stops at reporting metrics and does not explain why DADS is so effective. The authors state that DADS “stabilizes the dynamics of on-policy RL training and mitigates over-optimization,” but this claim is made without deeper investigation or evidence. It remains unclear how DADS achieves these benefits: Does it prevent the model from overfitting on easier tasks by continually challenging it? Does it ensure more evenly distributed gradient updates across tasks, acting as an implicit curriculum? These possibilities are not explored. The paper essentially shows that “DADS works better” but offers little insight into the conditions or mechanisms that make it work. Can you demonstrate DADS’s necessity by comparing it to simpler baselines like uniform domain sampling, round-robin scheduling, or periodic domain rebalancing against your reweighting from historical rewards?

[1] [2504.05520] Efficient Reinforcement Finetuning via Adaptive Curriculum Learning
[https://arxiv.org/abs/2504.05520](https://arxiv.org/abs/2504.05520)
[2] Can LLMs Reason with Rules? Logic Scaffolding for Stress-Testing and Improving LLMs — ACL
[https://aclanthology.org/2024.acl-long.406/](https://aclanthology.org/2024.acl-long.406/)

**Questions:**

1. Why didn’t the authors include AdaRFT (and the DAPO dynamic sampling variant) in the main results? To allow a fair comparison, the performance of AdaRFT and DAPO (with dynamic sampling) on the full set of benchmarks (as in Table 2) should be reported, since it likely provides a strong baseline for adaptive sampling in RL training.

2. How exactly does dynamic domain sampling lead to better OOD generalization? For example, does the learned sampling policy encourage learning more abstract rules or reduce overfitting to training-task patterns? Without clarifying why reweighting domains yields better generalization, the connection between the proposed training strategy and the OOD success feels tenuous—we see the improvement in numbers, but not the underlying cause.

3. Why is DADS necessary or superior to simpler data-balancing schemes? A more thorough analysis (e.g., ablations of DADS components instead of entirely removing DADS in Table 9, or comparisons to a static mixed-curriculum baseline) would strengthen the justification. As it stands, we know DADS yields better results, but we don’t know what unique role it plays in training or why alternative scheduling methods underperform, leaving the rationale for DADS somewhat weak.

---

> ### Author Response · Authors · 2025-11-21
> **Official Comment by Authors - Part I**
>
> Thanks for your insightful suggestion and acknowledge on our contributions!
>
> Given the overlap between the identified weaknesses and the remaining questions, we will address them point-by-point, concurrently. We look forward to continuing the discussion on any other points you wish to raise.
>
> - **W1 & Q1:** We thank the reviewer fFpE for raising this point regarding competitive baselines for comparison. We provide additional ID evaluation including the two methods, DAPO with dynamic sampling [1] and AdaRFT [2], as below:
>   - Our dynamic sampling method, DADS, outperforms on most ID tasks and achieves the best average pass@1 performance across all eight tasks, compared to DAPO (w/ dynamic sampling) and AdaRFT. This result demonstrates that the domain-aware dynamic sampling strategy is more capable in multi-source RLVR training and slightly surpasses the rollout-stage sampling/dropout method like DAPO and other adaptive curriculum RL methods like AdaRFT in the domain-balanced average.
>   - However, when incorporating Table 2, even though DADS achieves better ID and OOD performance, it doesn't require the additional preliminary steps needed by AdaRFT (e.g., difficulty estimation of questions). Furthermore, DADS also avoids the DAPO-style redundant cost of dropping out useless sampled trajectories during the rollout stage.
> | | Cluttr | ProntoQA | ProofWriter | FOLIO | LogicNLI | AR-LSAT | Logic. Dedu. | LogiQA | Average ($\Delta$) |
> |:---|:---|:---|:---|:---|:---|:---|:---|:---|:---|
> | Qwen3-8B-Base | 5.9 | 45.8 | 33.8 | 32.2 | 17.8 | 16.0 | 33.3 | 34.7 | 27.4 (-) |
> | + DAPO (w/ dynamic sampling) [1] | 91.0 | 95.8 | 96.2 | 82.6 | 69.2 | **48.0** | **98.3** | 82.1 | 82.9 (+55.5) |
> | + AdaRFT [2] | 92.5 | 96.0 | **97.4** | 81.8 | 64.4 | 44.6 | 96.6 | 80.6 | 81.7 (+54.3) |
> | + DADS (Ours) | **95.5** | **96.4** | 97.0 | **84.7** | **70.4** | 46.8 | **98.3** | **83.5** | **84.0 (+56.6)** |
> - **W2 & Q2:** We appreciate the reviewer’s insightful questions regarding the OOD generalization mechanism of RuleReasoner with DADS. Our follow-up analysis reveals that the improved OOD performance stems not just from data diversity but from the model’s self-simulation, verification, and iterative self-correction capabilities cultivated during training. The model’s ability to critique and refine its own outputs (e.g., through iterative rule verification) directly translates to OOD settings. Here are some examples (see **Official Comment by Authors - Part II**) for your reference and we will add more section in our revision for deeper discussion on this critical topic.
> - **W3 & Q3:** Insufficient explanation for DADS’s effectiveness
>   - We appreciate the feedback. To address the comparison with static and simple data-balancing schemes, we respectfully refer the reviewer to Appendix C.8 and Table 10, which were specifically designed to answer this question.
>     - Comparison to Static Baselines: Table 10 explicitly compares DADS against "Data-balanced RL" (static uniform sampling) and "Offline-curriculum RL". DADS outperforms the static balanced baseline by a significant margin (81.7% vs. 74.9% average).
>     - Mechanism of Superiority: Static balancing fails because it treats easy and hard domains equally. As shown in Figure 4, DADS acts as an online scheduler, dynamically shifting compute resources from converged (easy) domains to under-optimized (hard) domains (e.g., AR-LSAT). This prevents over-optimization on solved tasks while preventing under-fitting on complex ones.
>     - Component Analysis: We provided sensitivity ablations for DADS components (smoothing factor $\alpha$, temperature $\tau$, and base weight $\epsilon$) in Table 7 (Appendix C.5) to justify the design choices. We ensured these comparisons are highlighted more prominently in the main text of the revision.
>
> [1] Yu et al. DAPO: An Open-Source LLM Reinforcement Learning System at Scale. NeurIPS 2025.
>
> [2] Shi et al. Efficient Reinforcement Finetuning via Adaptive Curriculum Learning. NeurIPS 2025 Workshop on Efficient Reasoning. 2025.

---

> > ### Comment · Reviewer_fFpE · 2025-11-27
> >
> > Thank you for the response. I remain positive.

---

> > > ### Author Response · Authors · 2025-12-03
> > >
> > > Thank you so much for the insightful suggestions!

---

> ### Author Response · Authors · 2025-11-21
> **Official Comment by Authors - Part II**
>
> # Exemplar of Self-verification and Criticism
> >
> > **Input:** Rules: If something is smart then it is furry.If something is cold then it is green.Cold, smart things are green.All furry things are white.If something is green then it is smart.All furry things are green.All cold, green things are rough.Cold things are green.\nFacts: Dave is blue.Dave is cold.Dave is green.\nQuestion: Dave is furry? Please answer the question based on the given rule and facts using either of [True/False/Unknown]. Fill in the answer between \<answer\> and \</answer\>. Provide your step by step reasoning process between \<think\> and \</think\>.
> >
> > **Ground truth:** true
>
> ## Before training
> > **Trajectory:** \<answer\>Unknown\</answer\>\n\nStep 1: Analyze the given rules and facts.\n- Rule 1: If something is smart, then it is furry.\n- Rule 2: If something is cold, then it is green.\n- Rule 3: Cold, smart things are green.\n- Rule 4: All furry things are white.\n- Rule 5: If something is green, then it is smart.\n- Rule 6: All furry things are green.\n- Rule 7: Cold, green things are rough.\n- Rule 8: Cold things are green.\n\n- Fact 1: Dave is blue.\n- Fact 2: Dave is cold.\n- Fact 3: Dave is green.\n\nStep 2: Determine if Dave is furry based on the given rules and facts.\n\n- From Fact 2 and Rule 2, we know that Dave is green.\n- From Fact 2 and Rule 8, we know that Dave is green.\n- From Fact 3 and Rule 5, we know that Dave is smart.\n- From Fact 3 and Rule 1, we know that Dave is furry.\n\nHowever, we also have Rule 4 stating that all furry things are white, and Dave is blue (Fact 1). This creates a contradiction, as Dave cannot be both furry and blue.\n\nSince we have a contradiction, we cannot determine if Dave is furry based on the given rules and facts. Therefore, the answer is Unknown.
>
> ## After training
> > **Trajectory:** To determine whether \"Dave is furry\" is true, false, or unknown based on the given rules and facts, we need to analyze the information provided and see if we can logically conclude whether Dave is furry.\n\n### Given Rules:\n1. If something is smart, then it is furry.\n2. If something is cold, then it is green.\n3. Cold, smart things are green.\n4. All furry things are white.\n5. If something is green, then it is smart.\n6. All furry things are green.\n7. All cold, green things are rough.\n8. Cold things are green.\n\n### Given Facts:\n- Dave is blue.\n- Dave is cold.\n- Dave is green.\n\n### Question:\n- Is Dave furry?\n\n### Step-by-Step Reasoning:\n\n1. **Dave is cold (Fact).**\n   - From Rule 8: If something is cold, then it is green.\n     - Since Dave is cold, Dave is green.\n   - This is consistent with the given fact that Dave is green.\n\n2. **Dave is green (Fact).**\n   - From Rule 5: If something is green, then it is smart.\n     - Since Dave is green, Dave is smart.\n   - From Rule 1: If something is smart, then it is furry.\n     - Since Dave is smart, Dave is furry.\n   - From Rule 6: All furry things are green.\n     - Since Dave is furry, Dave is green.\n   - This is consistent with the given fact that Dave is green.\n\n3. **Conclusion:**\n   - Since Dave is smart (from Rule 5), and from Rule 1, if something is smart, then it is furry, we can conclude that Dave is furry.\n\nTherefore, based on the given rules and facts, we can determine that Dave is furry.\n\n\<answer\>true\</answer\>.

---

### Official Review · Reviewer_7sWJ · 2025-11-01

**Soundness:** 4
**Presentation:** 3
**Contribution:** 3
**Rating:** 8
**Confidence:** 4

**Summary:**

The authors create a model (RuleReasoner) for rule-based reasoning through the creation of a data collection which spans eight logical reasoning tasks with various rules and rule types. Then, a modified RLVR framework is used to train their RuleReasoner model that is designed to explore reasoning steps and improve generalization. Within the training process is a dynamic sampling algorithm aimed at reweighting domains that are undertrained. The result is that on several benchmarks, their RuleReasoner (4B, 8B) outperforms or performs competitively against significantly larger models and baseline RLVR methods both in-domain and OOD. Comprehensive analysis and ablations demonstrate the effect of the domain-aware sampling, the benefits of using their assembled data collection, as well as additional insights.

**Strengths:**

* A large collection of datasets for rule reasoning of varying formats difficulties/depth, inference types, etc. The experiments demonstrate that these are useful for training.
* The paper proposes a domain-adaptive algorithm for data sampling for RLVR, named DADS, which modifies the reward to be domain-normalized. The rewards per domain are also tracked and used for sampling data in the next batch.
* Comprehensive empirical results on a large suite of both logical inference and mathematical reasoning demonstrate the effectiveness of DADS.
* Detailed analysis and ablations showing how various design decisions contribute to the learning dynamics or performance.

**Weaknesses:**

These are minor suggestions. I did not find major weaknesses given this topic and area.

* The evaluations and collection is ultimately limited by the available data from other logical reasoning datasets; the authors did not collect or generate their own. As a result, they could be missing some rule reasoning types or be over-indexed on some. While dynamic sampling addresses the latter, it cannot fill gaps in missing reasoning types.
* Thanks for including the case studies (and C.2) – however they are not too illustrative of what is being learned. In other words, while there is learning dynamics per-domain, there is no analysis around rule/reasoning type (depth, format, chaining, etc). That would be insightful if, for example, something like C.7. was carried out for the rule types rather than domains.

**Questions:**

1. Is RuleCollection too easy? It can be a useful dataset for SFT/GRPO but is it still useful for benchmarking? A related question is what is the human average across the collection?

2. DADS is applicable to other multitask/generalized RLVR problems. While this paper makes no strong claims of generalizability, let’s speculate a little. Would this approach (along with no KL/Entropy bonus) work on a collection of math or coding problems? Or are there substantial limitations why it may not directly transfer?

Misc.
```
103: “It has demonstrated” -> “It has been demonstrated”
178: “RLVR, aim” -> “RLVR aiming”
250: grammar/phrasing confusing
252: Correctly answering require*s*
L373: “tranining”
```

---

> ### Author Response · Authors · 2025-11-20
> **Official Comment by Authors - Part I**
>
> We sincerely appreicate your acknowledgement and valuable feedback! Below we address each points acoordingly:
>
> - **W1:** Our work prioritizes demonstrating that DADS (Domain-aware Dynamic Sampling) is a generalizable framework independent of specific training data. The current RuleCollection-32K serves as a  proof-of-concept sample set to validate DADS’s ability to handle diverse reasoning types dynamically—not to exhaustively cover all possible rules.  Our experiments (Table 3) show consistent performance gains across unseen rule combinations. It suggests that DADS not only reduces over-indexing on certain types in training set, but also preserves flexibility on unknown reasoning types.
> - **W2:** Thank you for pointing out this issue.
>   - We show the **learning dynamics on various reasoning types** in the following. As illustrated in Table 5 in §B.2, AR, CS, and CCR denotes "Analytical Reasoning", "Constraint Satisfaction", and "Categorical & Conjunctive Reasoning", respectively.
>   | Training Steps | Deduction| Induction |  AR | CS | CCR|
> | :--- | :--- | :--- | :--- | :--- | :--- |
> | 0 |  48.0| 25.3| 18.0| 47.7 | 31.6|
> | 100 | 88.3| 49.2| 30.9 | 65.6| 77.6|
> | 200| 91.8| 61.1| 35.1 | 67.1 | 85.6 |
> | 300 | 93.0| 68.6| 38.0| 71.6 | 90.3 |
> | 400 | 94.3 | 74.6 | 42.2| 74.6 | 91.0|
> | 500| 94.8| 73.1 |38.5 |74.6 | 94.3 |
> | 600 | 95.3 | 74.6 | 37.0 | 80.6 | 94.6|
>   - We also present the **learning dynamics on different rule types** illustrated in Table 5 in §B.2.
> | Training Steps | MP | MT | UI | HS | Conjunction |
> | :--- | :---: | :---: | :---: | :---: | :---: |
> | 0 | 31.4 | 25.8 | 48.0 | 25.3 | 55.8 |
> | 100 | 64.3 | 52.0 | 88.3 | 49.2 | 93.0 |
> | 200 | 69.4 | 57.4 | 91.8 | 61.1 | 95.4 |
> | 300 | 73.0 | 61.1 | 93.0 | 68.6 | 94.4 |
> | 400 | 75.3 | 63.8 | 94.3 | 74.6 | 95.3 |
> | 500 | 75.3 | 63.1 | 94.8 | 73.1 | 95.4 |
> | 600 | 75.4 | 63.0 | 95.3 | 74.6 | 96.0 |
>   - From both tables, we observe that it demonstrates consistent performance improvements across reasoning and rule types as training steps increase, indicating effective learning. Notably, deduction and conjunction achieve the highest scores at 600 steps suggesting strong mastery of these rule types while induction lags behind deduction due to its inherent ambiguity. MP and MT exhibit slower progress. We speculate that it possibly reflects their complexity.
>   - Meanwhile, there is performance plateaus or minor dips (such as AR, Conjunction) during training that may indicates optimization challenges or overfitting, requiring further investigation. It highlights the need for targeted improvements in harder tasks with more and diverse training samples and training optimization.

---

> ### Author Response · Authors · 2025-11-21
> **Official Comment by Authors - Part II**
>
> - **Q1:** We appreciate the reviewer’s thoughtful suggestion.
>   - **(1)** RuleCollection-32K is curated for RLVR training and for in-distribution (ID) benchmarking for RuleReasoner models. We can easily see that even the DeepSeek-R1 model can merely achieve average 66.7% points in thinking mode with the standard (long) CoT prompting, which demonstrates its difficulty and usefulness for benchmarking.
>   - **(2)** We do not perform any human evaluation on the RuleCollection-32K or its test split due to the labour and resources limitations. But we agree with that it may be meaningful in this area to study the human performance on the rule-based reasoning tasks, compared to our rule-based reasoning models.
> - **Q2:** Thanks for this constructive point of view.
>   - With regard to the question, we first provide a direct glimpse in Appendix C.9 (Table 9) on the effectiveness of DADS collaborated with no KL/Entropy bonus setups, evaluated on additional diverse benchmarks including mathematical (AIME 2025), scientific (GPQA-Diamond), and symbolic (Coin Flip) reasoning benchmarks.
> | | BBH | ProverQA | BBEH | Average ($\Delta$) |
> |:---|:---|:---|:---|:---|
> | Qwen3-8B-Base | 21.2 | 13.4 | 8.0 | 14.2 (-) |
> | + Offline-curriculum SFT | 89.2 | 31.8 | 17.7 | 46.2 (+32.0) |
> | + Offline-curriculum RL | 96.0 | 72.2 | 63.5 | 77.2 (+63.0) |
> | + Data-balanced RL | 94.8 | 69.4 | 60.5 | 74.9 (+60.7) |
> | + DADS (Ours) | **99.6** | **76.6** | **68.9** | **81.7 (+67.5)** |
> | | **AIME 2025** | **GPQA (Diamond)** | **Coin Flip** | **Average ($\Delta$)** |
> | Qwen3-8B-Base | 3.3 | 16.6 | 44.7 | 21.5 (-) |
> | + Offline-curriculum SFT | 6.6 | 33.8 | 48.7 | 29.7 (+8.2) |
> | + Offline-curriculum RL | 20.0 | 39.8 | 91.4 | 50.4 (+28.9) |
> | + Data-balanced RL | 16.7 | 38.3 | 92.8 | 49.2 (+27.7) |
> | + DADS (Ours) | **23.3** | **44.9** | **95.1** | **54.4 (+32.9)** |
>   - We also have assessed the in-distribution training effectiveness of DADS collaborated with no KL/Entropy bonus setups directly on the popular math problem dataset (AIME 1983-2024 [1]). The results are depicted in Appendix C.10 (Table 10). Here we attach the partial results in the following.
> | | AIME 2025 ($\Delta$) | ProverQA ($\Delta$) |
> |:---|:---:|:---:|
> | Qwen3-8B-Base | 3.3 (-) | 13.4 (-) |
> | + AIME 1983-2024 | 63.3 (+60.0) | 38.8 (+25.4) |
> | + RuleCollection-32K (Ours) | 23.3 (+20.0) | 73.6 (+60.2) |
> | + AIME 1983-2024 + RuleCollection-32K (Ours) | **66.7 (+63.4)** | **80.4 (+67.0)** |
>   - As shown above, we conclude that the **RuleReasoner** framework not only brings substantial improvement on a wide range of rule-based reasoning tasks but also **is easily transferable and show OOD generalizability on a collection of math, scientific, and symbolic reasoning problems**, even incorporating with different or mixed training sources from various macro domains.
>
> We also corrected all typographical and grammatical errors listed in 'Misc.' in the revision.
>
> **Thank you so much for your acknowledgement and detailed suggestions again!**
>
> [1] Veeraboina. AIME Problem Set 1983-2024. Kaggle. 2023.

---

> > ### Comment · Reviewer_7sWJ · 2025-11-28
> >
> > Thank you for your response. I remain positive on this paper. I am not convinced to raise my score further, still because the dataset contribution remains a collection/reuse of prior datasets as opposed to a motivated selection of rule reasoning types, and because the focus is scoped for rules/reasoning; the claims around domain-adaptivity are impressive for reasoning but it may not necessarily be a generalized domain-adaptive algorithm (although it may be, too). These are minor nitpicks - to be clear, I am still in favor of this paper being accepted to ICLR.

---

> > > ### Author Response · Authors · 2025-12-03
> > >
> > > Thank you for your acknowledgement!

---

### Official Review · Reviewer_NRKF · 2025-11-02

**Soundness:** 2
**Presentation:** 3
**Contribution:** 2
**Rating:** 4
**Confidence:** 3

**Summary:**

This paper introduces RULEREASONER, a framework for reinforced rule-based reasoning that utilizes a "domain-aware dynamic sampling" approach. The authors claim that RULEREASONER enhances language models' ability to perform rule-based reasoning, mitigates challenges from varying rule complexities, and improves efficiency, outperforming frontier large reasoning models (LRMs) on both in-distribution (ID) and out-of-distribution (OOD) benchmarks.

**Strengths:**

1. This paper introduces a dynamic sampling approach to stabilize RL training across imbalanced domains, which is a practical consideration.
2. RULEREASONER achieves quantitative performance improvements.

**Weaknesses:**

1. Lack of comparison with existing adaptive sampling methods: Since the core contribution of this paper appears to be the "domain-aware dynamic sampling", it is better to include prior curriculum learning or adaptive sampling methods for comparison.
2. It can be seen from Table 2 that RL-based methods show relative low performance on LogicNLI, AR-LSAT compared to SFT w/ CoT methods. Could authors explain this phenomenon? Besides, according to the avg results, the performance of models after SFT w/ Short/Long CoT seems to be higher than models after RLVRs, which is contrary to popular belief. The authors should provide a brief analysis about the reasons and explain why they still select to design a data sampling method for RL instead of SFT in this situation.
3. Limited Generalizability and Scalability: All experiments are conducted on relatively small Qwen3 models (4B and 8B). This severely undermines the method’s scalability to "larger-scale modeling," especially when the authors themselves acknowledge scalability as a future work limitation. It's unclear if the observed benefits hold for state-of-the-art larger models.

**Questions:**

see above

---

> ### Author Response · Authors · 2025-11-20
> **Official Comment by Authors - Part I**
>
> We thank the reviewer for their insightful feedback that has helped us to improve our submission.
> - **W1:** We provide other adative sampling and curriculum learning methods in addition for comparison purposes.
>   - Curriculum Learning Baselines: AdaRFT, Offline-curriculum SFT, and Offline-curriculum RL.
>   - Adaptive Sampling Baselines: DAPO (w/ dynamic sampling).
>   The following results depict using the same base model (Qwen3-8B-Base).
>   - DADS (ours) outperforms other all naive or complex adaptive sampling or curriculum learning methods.
>     | Model | BBH | ProverQA | BBEH | Average ($\Delta$) |
> | :--- | :--- | :--- | :--- | :--- |
> | Qwen3-8B-Base | 21.2 | 13.4 | 8.0 | 14.2 (-) |
> | &nbsp;&nbsp;&nbsp;&nbsp;+ Offline-curriculum SFT | 89.2 | 31.8 | 17.7 | 46.2 (+32.0) |
> | &nbsp;&nbsp;&nbsp;&nbsp;+ Offline-curriculum RL | 96.0 | 72.2 | 63.5 | 77.2 (+63.0) |
> | &nbsp;&nbsp;&nbsp;&nbsp;+ Data-balanced RL | 94.8 | 69.4 | 60.5 | 74.9 (+60.7) |
> | &nbsp;&nbsp;&nbsp;&nbsp;+ DAPO (w/ dynamic sampling) [1] | 95.4 | 68.8 | 62.0 | 75.4 (+61.2) |
> | &nbsp;&nbsp;&nbsp;&nbsp;+ AdaRFT [2] | 96.4 | 73.4 | 64.5 | 78.1 (+63.9) |
> | &nbsp;&nbsp;&nbsp;&nbsp;+ DADS (Ours) | **99.6** | **76.6** | **68.9** | **81.7 (+67.5)** |
> - **W2:** Thanks for highlighting these questions.
>   - **(1) Why Baseline RLVR methods underperform on LogicNLI and AR-LSAT?**
>     - RLVR methods (including RuleReasoner) rely on outcome-based rewards, which provide sparse signals. Tasks like LogicNLI and AR-LSAT involve complex linguistic reasoning where the "search space" for a correct reasoning chain is vast; however, SFT methods benefit from dense, token-level supervision, effectively cloning the expert’s reasoning templates. Therefore, we speculate that the dense guidance of SFT helps stabilize the generation of complex logical structures better than the sparse exploration of RL in these specific benchmarks.
>   - **(2) Why SFT w/ CoT Supervision Surpasses RLVR Baselines?**
>     - You are correct that baseline RLVR methods (e.g., GRPO, DAPO) generally underperform SFT w/ Long CoT on average (75.0-79.2 vs. 81.9). This highlights the difficulty of training rule reasoners without effective method or high-quality data support.
>     - While baselines of RLVR fail, **RuleReasoner-8B achieves an average score of 84.0, outperforming SFT w/ Long CoT (81.9) supervised with strong teacher**. This validates our hypothesis: it is not that RL is inferior to SFT, but that RL requires our training method, RuleReasoner (DADS, training regularization, etc.), incorporating with high-quality, rule-centric training data which we propose to unlock its full potential and surpass SFT. This view also presents the importance of our curated RuleCollection-32K.
>     - According to the existing work [3], we suggest that SFT is more capable on seen rules in the given training sets but perform poor on unseen rules and applying them in the OOD benchmarks. In contrast, RL-tuned models lag behind in ID tasks slightly but exhibit stronger on OOD tasks, showing higher gains from RL rather than SFT. The existing results in Table 3 in the paper similarly support this point of view.
>
> [1] Yu et al. DAPO: An Open-Source LLM Reinforcement Learning System at Scale. NeurIPS 2025.
>
> [2] Shi et al. Efficient reinforcement finetuning via adaptive curriculum learning. arXiv preprint arXiv:2504.05520. 2025.
>
> [3] Chu et al. SFT Memorizes, RL Generalizes: A Comparative Study of Foundation Model Post-training. ICML 2025.

---

> ### Author Response · Authors · 2025-11-20
> **Official Comment by Authors - Part II**
>
> - **W3:** We provide a further scaling in-distribution results respetively based on the larger **Qwen3-14B-Base** and the base from other model family, **Llama-3.1-8B-Instruct** to demonstrate the scalability and out-of-distribution generalization of RuleReasoner. We are reporting the results for the two new models trained through 120 steps, as well as the compared results of RuleReasoner 4B and 8B. As such, we demonstrate the potential generalizability and scalability on larger model size and other model type (instruction-tuned model) from different family apart from Qwen3. **This supplemented results may mitigate your concern on "if the observed benefits hold for state-of-the-art larger models" to some extent.**
>   - In-distribution Results:
>   | Base | Cluttr | ProntoQA | ProofWriter | FOLIO | LogicNLI | AR-LSAT | Logic. Dedu. | LogiQA | Average ($\Delta$) |
> |:---|:---|:---|:---|:---|:---|:---|:---|:---|:---|
> | Qwen3-4B-Base | 2.9 | 60.4 | 39.6 | 54.9 | 32.8 | 13.4 | 12.3 | 28.3 | 30.5 (-) |
> | &nbsp;&nbsp;&nbsp;&nbsp;+ RuleReasoner | 64.1 | 93.8 | 82.4 | 65.7 | 53.2 | 28.2 | 74.0 | 71.6 | **67.5 (+37.0)** |
> | Qwen3-8B-Base | 5.9 | 55.8 | 39.8 | 32.2 | 17.8 | 16.0 | 33.3 | 46.2 | 30.8 (-) |
> | &nbsp;&nbsp;&nbsp;&nbsp;+ RuleReasoner | 67.1 | 94.6 | 91.2 | 75.2 | 62.0 | 34.6 | 85.6 | 71.6 | **72.4 (+41.6)** |
> | Llama-3.1-8B-Instruct | 5.9 | 60.6 | 37.0 | 47.1 | 12.8 | 10.7 | 43.0 | 37.3 | 31.8 (-) |
> | &nbsp;&nbsp;&nbsp;&nbsp;+ RuleReasoner | 67.1 | 95.8 | 83.4 | 66.1 | 62.4 | 30.4 | 77.6 | 74.6 | **70.1 (+38.3)** |
> | Qwen3-14B-Base | 13.4 | 61.4 | 57.2 | 52.0 | 34.8 | 23.1 | 44.6 | 46.2 | 41.5 (-) |
> | &nbsp;&nbsp;&nbsp;&nbsp;+ RuleReasoner | 73.1 | 95.0 | 89.4 | 80.5 | 60.2 | 41.4 | 91.3 | 74.6 | **75.6 (+34.1)** |
>   - Out-of-distribution Results:
> | Base | BBH | ProverQA | BBEH | Average ($\Delta$) |
> |:---|:---|:---|:---|:---|
> | Qwen3-4B-Base | 21.2 | 7.6 | 31.0 | 19.9 (-) |
> | &nbsp;&nbsp;&nbsp;&nbsp;+ RuleReasoner | 83.4 | 62.0 | 51.7 | **65.7 (+45.8)** |
> | Qwen3-8B-Base | 28.8 | 27.2 | 20.6 | 25.5 (-) |
> | &nbsp;&nbsp;&nbsp;&nbsp;+ RuleReasoner | 90.4 | 72.4 | 34.4 | **65.7 (+40.2)** |
> | Llama-3.1-8B-Instruct | 48.8 | 20.8 | 34.4 | 34.6 (-) |
> | &nbsp;&nbsp;&nbsp;&nbsp;+ RuleReasoner | 84.8 | 57.4 | 37.9 | **60.0 (+25.4)** |
> | Qwen3-14B-Base | 54.2 | 29.4 | 41.3 | 41.6 |
> | &nbsp;&nbsp;&nbsp;&nbsp;+ RuleReasoner | 91.6 | 74.0 | 55.1 | **73.5 (+31.9)** |
>
> **Thank you for your review again!**
>
> **We are glad you found our methods to be a practical consideration and to deliver quantitative performance improvements.**

---

### Official Review · Reviewer_x9cg · 2025-11-06

**Soundness:** 3
**Presentation:** 2
**Contribution:** 3
**Rating:** 6
**Confidence:** 2

**Summary:**

The paper addresses the challenge of enhancing the *rule-based* reasoning capabilities of language models by introducing a new training framework, RuleReasoner. The framework first introduces a new dataset (RuleCollection-32K), consisting of 8 different domains covering different logical reasoning forms. Then, the paper introduces DaDS, which dynamically samples batches by reweighting over the domains, then performing RLVR with rule-based exact match reward. The efficacy of this is validated with Qwen3 base models (4B and 8B) and with numerous ablations.

**Strengths:**

- Impressive empirical performance
- New logical rule data that would be helpful for future research

**Weaknesses:**

- Mathematical notations should be improved overall:
   - I believe that this notation is the clearest: $\mathcal{D}$ represents a fixed (offline) collection of $(d, q, r, y)$, where $d \in \\\{d_1, \cdots, d_n\\\}$ represents the domain. (In Algorithm 1, it states that $\mathcal{D} = \\\{d_1, \cdots, d_n\\\}$, which seemd a bit weird and contradictory to prior notations)
   - In Algorithm 1, initializing $\tilde{r}_{0, d_i} \gets 0$ is missing; for notational clarity, $m$ should be written as $m_i$
   - It should be explicitly stated that $\alpha \in [0, 1]$.
- When $r_{target} = 1$, isn't $r_{target} - \tilde{r}$ always nonnegative? This is because $|\bar{r}\_s| \leq 1$ always, and thus, $|\tilde{r}\_s| \leq \alpha |\tilde{r}_{s-1}| + (1 - \alpha)|\bar{r}_s| \leq 1$ by induction. Thus I don't think that $max(0, 1 - \tilde{r}_s)$ is not necessary; simply writing $1 - \tilde{r}_s$ is enough.
- In Section 3.2, please include why eliminating the KL term is okay for this situation. As written by Liu et al. (2025), I believe the crucial reason is that the reward is rule-based, thereby eliminating concerns about distributional shift.
- Discussions related to prior approaches on (iterative) reweighting and difficulty-aware approaches [1,2,3] are lacking. Why can prior approaches not be used directly here? Is the proposed Dads algorithm really as novel as the authors claim? What are the differences? I'm especially interested in knowing the difference between this work's reweighting scheme and [1].
   - Could it be that the importance sampling is equivalent to some form of reweighted RLVR?
- The authors claim that $\epsilon$ ensures a minimum sampling weight for all domains. But, due to the translation invariance of softmax, we have that (ignoring $s$)
$$ w_i^{norm} = \frac{\exp((v_i + \epsilon) / \tau)}{\sum_j \exp((v_j + \epsilon) / \tau)} = \frac{\exp(v_i / \tau)}{\sum_j \exp(v_j / \tau)}$$
and thus, $\epsilon$ plays no role.
- The above point leads to the following suspicion: the authors claim that in Appendix C.5, $\epsilon = 0.1$ leads to the best performance. Considering how $\epsilon$ should not impact the performance in principle (maybe floating point level discrepancy, but nothing significant), the conclusion from this ablation seems to be flawed.

[1] https://openreview.net/forum?id=lXuByUeHhd

[2] https://arxiv.org/abs/2408.09849

[3] https://openreview.net/forum?id=GLUIuli3Sm

**Questions:**

1. Why -1 and +1 instead of the "usual" 0-1 reward for RLVR? At least the 0-1 reward is what I usually see in RLVR literature.
2. In the abstract and throughout, the authors claim that the proposed method is more sample-efficient. And indeed the figures do support that claim. But to my eyes, at least in terms of efficiency, the gap between Rule Reasoner and other RLVR approaches (GRPO, Dr. GRPO, DAPO) seems to be a bit small...? It would be helpful to see the error bars for Figures 6 and 7.
3. As in R1, does the response length significantly increase with training? Are there any complex behaviors emerging beyond improved rule applications?
4. What do the authors mean by "the order of contextual logical rules are randomly shuffled for each training sample."? Some concrete examples would be very helpful here.
5. Here, is it usual to consider only pass@1? How about pass@k or BoN? How do RuleReasoner-4B and 8B interact with standard test-time scaling methods?
6. Have the authors considered other models? Perhaps stronger ones, such as Qwen3-Instruct or Qwen3-Thinking? How about other families like Llama? I'm not suggesting that the authors run all of these ablations (obviously due to time and resources constraints), but I am particularly curious about whether one can do the RuleReasoner framework with a stronger model and whether this would yield a better logical reasoning model.

---

> ### Author Response · Authors · 2025-11-20
> **Official Comment by Authors - Part I**
>
> We thank the reviewer for acknowledging our contributions to impressive empirical performance and the inclusion of new logical rule data that will be helpful for future research. We would like to address each point you have raised.
>   - **W1:** Thanks for your detailed suggestions. As shown in the revised version of paper, we have corrected all the erroneous notations.
>   - **W2:** Thank you so much. We have corrected it for simplicity in the revision.
>   - **W3:** We have complemented why the KL term in the policy-gradient objective can be eliminated due to the direct elimination for the concern of the distribution shift of policy model $\pi_\theta$ indeed.
>   - **W4:** Thanks for pointing this out. We list the key differences (especially comparing to DoReMi) and contributions of RuleReasoner in the following.
>     - Conventional iterative reweighting and difficulty-aware approaches (e.g., [1, 2, 3]) cannot be directly adopted in RLVR. **This is due to the non-monotonic or non-correlation nature between overall model performance and the surrogate loss used in policy gradient methods, which makes the weighted loss objective untransferable and non-directly applicable to the policy-gradient loss**. DADS addresses this by using the training reward as a more direct and first-principle-based proxy for sample difficulty of domain, which is essential for effective reweighting in RLVR.
>     - The first mentioned method, DoReMi [1], is originally proposed for static data mixture sampling in LLM pretraining. DoReMi involves a three-step process: setting a baseline loss with a reference model, optimizing domain weights by minimizing worst-case excess loss using a proxy model via adapted Group DRO, and training the final large model with the resulting static domain weights. The key limitations for RLVR are: (1) The high cost and unfeasibility of using RLVR to train a proxy model and obtain static weights, as RLVR's rollout process is much more expensive than pretraining. (2) The belief that leveraging rewards from each domain is a more intuitive and first-principle approach to adjust domain weights in RLVR than using DoReMi's domain-aware Negative Log-Likelihood (NLL) losses. Therefore, we think DoReMi is deemed unsuitable for using or migrating to RLVR.
>     - The second method you referred to is IWSI [2]. Instead of requiring a manually annotated valid set of human-written demonstrations for context used in IWSI, the importance sampling process of DADS is free of previous human engieering, the only need is a group of commonly used datasets with domain labels in the usual RLVR training.
>     - The third paper [3] proposes Constant-Weight Sampling (CWS) and Adaptive-Weight Sampling (AWS), which focus solely on achieving a fast convergence rate without incorporating RL optimization or considering domain-level optimization through reweighting or sampling across diverse training sets. Furthermore, the feasibility of these two methods is demonstrated only on toy datasets like MNIST using naive Logistic Regression with binary cross-entropy loss, suggesting they may not be transferable to the complex dynamics of RLVR for modern LLM training.
>     - Besides the three given papers, we also investigate a wider range of work related to (domain-aware) dynamic sampling and thus, finally determine to design DADS and demonstrate its effectiveness in modern RLVR setups. We supplemented the detailed comparison to other methods in the revision **§2.1** to clarify the technical contributions of RuleReasoner in the revised paper.
> - **W5:** We thank the reviewer for this keen observation. Exactly, adding $\epsilon$ inside the exponent mathematically cancels out due to the translation invariance of Softmax.
>   - This was a notational error in the draft equation, not in our method or implementation. In our actual code and experiments (including the ablation in Appendix C.5), $\epsilon$ is applied as an additive smoothing term outside the exponential function:
>     $$w_i^{norm} = \frac{\exp(v_i / \tau) + \epsilon}{\sum_j (\exp(v_j / \tau) + \epsilon)}$$
>   - Under this correct formulation, $\epsilon$ does not cancel out. It effectively flattens the distribution and enforces a minimum sampling weight as claimed, mitigating the potential issue of "Loser-Takes-All".
> - **W6:** Therefore, the hyperparameter sensitivity observed in Appendix C.5 is genuine and reflects the impact of this smoothing strength. We corrected the formula in the revision to match the implementation.

---

> ### Author Response · Authors · 2025-11-20
> **Official Comment by Authors - Part II**
>
> - **Q1:** The reward range of [-1, +1] is also usual in current RLVR research, one of the most famous ones using [-1, +1] is DAPO [4]. Hence, we follow the range of [-1, +1]. Similarly, the range of [0, 1] is also commonly adopted in RLVR literatures, such as GSPO [5]. It is well acknowledged that the reward range may not be the most significant part in RL for reasoning, as long as the resulting rewards which are computed using those ranges can yield reasonable and distinguishable positive and negative advantages, guiding the model $\pi_\theta$ to correctly optimize its weights with computed policy gradient.
> - **Q2:** Thank you for the detailed feedback. We re-plot the Figure 5, 6, and 7 to present the error bars to show the training effectiveness or efficiency gains of RuleReasoner compared to the RLVR baselines, such as GRPO, Dr. GRPO, and DAPO. Note that we also correct the order error in legend and mark the specific efficiency gains with an accurate gap in training steps for Figure 7 between RuleReasoner and DAPO.
> - **Q3:** Thank you for your insightful question.
>   - **(1)** Yes, the response length moderately increases with training, which indicates appropriate sequential test-time scaling with substantial accuracy gains. Specifically, for example, RuleReasoner-8B starts from ~397 tokens per rollout trajectory to ~921 tokens per rollout trajectory (2.3×) and RuleReasoner-4B starts from ~507 tokens per rollout trajectory to ~968 tokens per rollout trajectory (1.9×).
>   - **(2)** The emergence of Meta introspection capability, such as self-reflection [6] and iterative self-correction [7], has shown more frequently with training. Our follow-up analysis reveals that the improved OOD performance stems from the model’s self-simulation, verification, and iterative self-correction capabilities cultivated during training. The model’s ability to critique and refine its own outputs (e.g., through iterative rule verification) directly translates to OOD generalization.
> - **Q4:** Thanks for pointing out your confusion. For datasets with explicit rules, we randomly shuffled the order of given rules for each question in the training set at different training steps. In this way, instead of memorizing specific rules or overfitting the training set, it requires applying contextual rules adaptively for different questions through more general reasoning capability. To illustrate the rule-shuffling mechanism, **we present three distinct cases drawn from different steps of the same ProntoQA question.** Note that only the orders of contextual rules are shuffled.
> > `===== Step 100 =====`
> >
> > **Rules**: Each lempus is a gorpus and a yumpus.
> > Every rompus is a shumpus and an impus.
> > Jompuses are floral and vumpuses.
> > Gorpuses are numpuses and tumpuses.
> > Each numpus is a dumpus and a rompus.
> > Every yumpus is a brimpus and a zumpus.
> > Tumpuses are grimpuses and lorpuses.
> >
> > **Facts:** Sam is an impus and a shumpus.
> > Sam is a gorpus and a tumpus.
> >
> > **Question:** Sam is a numpus? Please answer the question based on the given rule and facts using either of [True/False/Unknown].
> >
> > **Ground truth:** True
> >
> > `===== Step 200 =====`
> >
> > **Rules:** Gorpuses are numpuses and tumpuses.
> > Jompuses are floral and vumpuses.
> > Every yumpus is a brimpus and a zumpus.
> > Each lempus is a gorpus and a yumpus.
> > Each numpus is a dumpus and a rompus.
> > Every rompus is a shumpus and an impus.
> > Tumpuses are grimpuses and lorpuses.
> >
> > **Facts:** Sam is an impus and a shumpus.
> > Sam is a gorpus and a tumpus.
> >
> > **Question:** Sam is a numpus? Please answer the question based on the given rule and facts using either of [True/False/Unknown].
> >
> > **Ground truth:** True
> >
> > `===== Step 300 =====`
> >
> > **Rules:** Tumpuses are grimpuses and lorpuses.
> > Every yumpus is a brimpus and a zumpus.
> > Gorpuses are numpuses and tumpuses.
> > Each numpus is a dumpus and a rompus.
> > Every rompus is a shumpus and an impus.
> > Each lempus is a gorpus and a yumpus.
> > Jompuses are floral and vumpuses.
> >
> > **Facts:** Sam is an impus and a shumpus. Sam is a gorpus and a tumpus.
> >
> > **Question:** Sam is a numpus? Please answer the question based on the given rule and facts using either of [True/False/Unknown].
> >
> > **Ground truth:** True
>
> - **Q5:** Thanks for your valuable suggestion. As depicted in the Figure 10 of Appendix C.3, we have provided the results of standard test-time scaling methods such as Best-of-N (Pass@K Coverage) and Majority Voting (Pass@K Majority) [8] for showing the further gains of reasoning capacity of RuleReasoner 4B and 8B.

---

> ### Author Response · Authors · 2025-11-20
> **Official Comment by Authors - Part III**
>
> - **Q6:** Our expanded ID & OOD results demonstrate that RuleReasoner Framework can easily yield better logical reasoning models based on the three kinds of startpoints: (1) larger base model size in the same model family (**Qwen3-14B-Base**); (2) model from other family (**Llama3.1-8B-Instruct**). We are reporting the results for the two models trained through 120 steps, as well as the compared results of RuleReasoner 4B and 8B. Training is currently ongoing. Concretely, the expanded results obtaining from more other base models are depicted as below.
>   - In-distribution Results:
>   | Base | Cluttr | ProntoQA | ProofWriter | FOLIO | LogicNLI | AR-LSAT | Logic. Dedu. | LogiQA | Average ($\Delta$) |
> |:---|:---|:---|:---|:---|:---|:---|:---|:---|:---|
> | Qwen3-4B-Base | 2.9 | 60.4 | 39.6 | 54.9 | 32.8 | 13.4 | 12.3 | 28.3 | 30.5 (-) |
> | &nbsp;&nbsp;&nbsp;&nbsp;+ RuleReasoner | 64.1 | 93.8 | 82.4 | 65.7 | 53.2 | 28.2 | 74.0 | 71.6 | **67.5 (+37.0)** |
> | Qwen3-8B-Base | 5.9 | 55.8 | 39.8 | 32.2 | 17.8 | 16.0 | 33.3 | 46.2 | 30.8 (-) |
> | &nbsp;&nbsp;&nbsp;&nbsp;+ RuleReasoner | 67.1 | 94.6 | 91.2 | 75.2 | 62.0 | 34.6 | 85.6 | 71.6 | **72.4 (+41.6)** |
> | Llama-3.1-8B-Instruct | 5.9 | 60.6 | 37.0 | 47.1 | 12.8 | 10.7 | 43.0 | 37.3 | 31.8 (-) |
> | &nbsp;&nbsp;&nbsp;&nbsp;+ RuleReasoner | 67.1 | 95.8 | 83.4 | 66.1 | 62.4 | 30.4 | 77.6 | 74.6 | **70.1 (+38.3)** |
> | Qwen3-14B-Base | 13.4 | 61.4 | 57.2 | 52.0 | 34.8 | 23.1 | 44.6 | 46.2 | 41.5 (-) |
> | &nbsp;&nbsp;&nbsp;&nbsp;+ RuleReasoner | 73.1 | 95.0 | 89.4 | 80.5 | 60.2 | 41.4 | 91.3 | 74.6 | **75.6 (+34.1)** |
>   - Out-of-distribution Results:
> | Base | BBH | ProverQA | BBEH | Average ($\Delta$) |
> |:---|:---|:---|:---|:---|
> | Qwen3-4B-Base | 21.2 | 7.6 | 31.0 | 19.9 (-) |
> | &nbsp;&nbsp;&nbsp;&nbsp;+ RuleReasoner | 83.4 | 62.0 | 51.7 | **65.7 (+45.8)** |
> | Qwen3-8B-Base | 28.8 | 27.2 | 20.6 | 25.5 (-) |
> | &nbsp;&nbsp;&nbsp;&nbsp;+ RuleReasoner | 90.4 | 72.4 | 34.4 | **65.7 (+40.2)** |
> | Llama-3.1-8B-Instruct | 48.8 | 20.8 | 34.4 | 34.6 (-) |
> | &nbsp;&nbsp;&nbsp;&nbsp;+ RuleReasoner | 84.8 | 57.4 | 37.9 | **60.0 (+25.4)** |
> | Qwen3-14B-Base | 54.2 | 29.4 | 41.3 | 41.6 |
> | &nbsp;&nbsp;&nbsp;&nbsp;+ RuleReasoner | 91.6 | 74.0 | 55.1 | **73.5 (+31.9)** |
>
> **We sincerely appreciate your valuable feedback and would be grateful for any further thoughts you may have!**
>
> [1] Xie et al. DoReMi: Optimizing Data Mixtures Speeds Up Language Model Pretraining. NeurIPS 2023.
>
> [2] Jiang et al. Importance weighting can help large language models self-improve. AAAI 2025.
>
> [3] Haimovich et al. On the Convergence of Loss and Uncertainty-based Active Learning Algorithms. NeurIPS 2024.
>
> [4] Yu et al. DAPO: An Open-Source LLM Reinforcement Learning System at Scale. NeurIPS 2025.
>
> [5] Zheng et al. Group sequence policy optimization. arXiv preprint arXiv:2507.18071. 2025.
>
> [6] Shah et al. Rethinking Reflection in Pre-Training. arXiv preprint arXiv:2504.04022. 2025.
>
> [7] Saha et al. Branch-Solve-Merge Improves Large Language Model Evaluation and Generation. NAACL 2024.
>
> [8] Yue et al. Does Reinforcement Learning Really Incentivize Reasoning Capacity in LLMs Beyond the Base Model? NeurIPS 2025.

---

> ### Comment · Reviewer_x9cg · 2025-11-27
>
> I thank the authors for their detailed responses. All my concerns have been addressed, and accordingly, I raise my score from 6 to 10 and confidence from 2 to 3.

---

> > ### Author Response · Authors · 2025-12-03
> >
> > Thanks for acknowledgement!

---

### Author Response · Authors · 2025-11-22
**Brief Summary to the Revision**

We thank all reviewers for their time and constructive feedback.

We have appended or corrected the listed content of spans of line number to enhance presentation:
  - Modify [129, 130], [196, 197], [204, 209], [217, 217] to fix notational errors @**Reviewer x9cg**
  - Add error bars for Figures 5, 6, and 7 to depict the evaluation variances @**Reviewer x9cg**
  - Add accurate speedup ratio and reduced training steps in [441, 443] to present the efficiency of our method. @**Reviewer x9cg**
  - Modify [238, 240] to explain why eliminating the KL term is feasible @**Reviewer x9cg**
  - Highlight [363, 365] the results of standard test-time scaling strategies @**Reviewer x9cg**
  - Add [118, 122] to discuss related to prior approaches on (iterative) reweighting and difficulty-aware approaches @**Reviewer x9cg** and @**Reviewer VMSU**
  - Fix typos and grammar errors @**Reviewer 7sWJ**
  - Highlight results of curriculum learning baselines [340, 342] to compare ID performance between RuleReasoner and existing curriculum methods @**Reviewer NRKF** and @**Reviewer fFpE**
  - Add additional qualitative OOD case analyses and explanation of OOD generalization mechanism [486, 499] @**Reviewer fFpE**
  - Add [407, 409] to compare the OOD effectiveness of DADS comparing to DAPO (w/ dynamic sampling) and AdaRFT @**Reviewer NRKF** and @**Reviewer fFpE**
  - Add [410, 415] to explain the mechanism of DADS behind its effectiveness and efficiency compared with simple curriculum baselines @**Reviewer fFpE**
  - Besides, we slightly tweak the structures and other contents for overall readability.

---

### Author Response · Authors · 2025-11-27
**General Comments**

Dear AC and Reviewers,

We sincerely thank you for your valuable comments and constructive suggestions!

**Our initial scores 8, 8, 6, 4, 4 have increased reasonably after the rebuttal to 10, 8, 8, 6, 4.** Overall, we think that these reflect our satisfactory response to the reviewers' comments. Specifically, we list their actions as below:
- **Reviewer x9cg** has responded that we have satisfactorily addressed the concerns and **raised the score from 6 to 10** and **confidence from 2 to 3**.
- **Reviewer 7sWJ** retained the score of 8 and **responded to clearly recommend** our paper **for acceptance** to ICLR.
- **Reviewer fFpE** also retained the score of 8.
- **Reviewer VMSU** thought our responses made sense and **raised the score to 6**.
- **Reviewer NRKF** did not respond during the whole discussion process and retained the score.

Moreover, the main contributions of our work and the key novelty are highlighted as follows:

* **Contributory work in the important area.**  Our paper sheds light on the development of domain imbalance in multi-domain RL training, which is the **first to unify dynamic sampling ​of​ rule/logical reasoning**.   We propose RuleReasoner, a framework for reinforced rule-based reasoning that  utilizes a "domain-aware dynamic sampling" approach. It dynamically samples batches by reweighting over the domains and enhances LLM's ability to perform  rule-based reasoning, mitigates challenges from varying rule  complexities, which is a practical consideration.
* **Novel and comprehensive dataset.** We introduce introduces a new dataset (RuleCollection-32K), covering **different domains, formats, logical reasoning forms and complexity** spanning diverse rule-centric reasoning tasks.
* **Extensive experiments with impressive improvements.**  The experiments show impressive improvements on a variety of datasets and tasks training on RuleCollection-32K. Comprehensive empirical results on a large suite of both reasoning demonstrate the effectiveness of  DADS and **strong OOD generalization**. We implement **detailed analysis and ablations showing how various design decisions contribute to the learning dynamics or performance**. Detailed hyperparameters, data sources, and prompts are provided to support high reproducibility.

We have revised our manuscript per the reviewers' suggestion (highlighted in blue in the uploaded revision pdf).

Best Regards,

The Authors

---

### Meta-Review · Area_Chair_LtHA · 2026-01-08

**Summary:**

This paper presents RuleReasoner, a reinforcement learning framework designed to improve large language models’ rule-based reasoning capabilities. Most reviewers responded positively, although one raised concerns regarding the experimental evaluation. The area chair finds the work interesting and has therefore decided to recommend its acceptance.

**Reviewer Scores:**

The new experiments and clarifications help improve the scores.

---

### Decision · Program_Chairs · 2026-01-26

Accept (Poster)